# The role of PDF neurons in setting the preferred temperature before dawn in *Drosophila*

Xin Tang[1], Sanne Roessingh[2,3], Sean E Hayley[1], Michelle L Chu[1],
Nobuaki K Tanaka[4,5], Werner Wolfgang[3], Seongho Song[6], Ralf Stanewsky[2†],
Fumika N Hamada[1,5,7,8*]

[1]Visual Systems Group, Abrahamson Pediatric Eye Institute, Division of Pediatric Ophthalmology, Cincinnati Children's Hospital Medical Center, Cincinnati, United States; [2]Department of Cell and Developmental Biology, University College London, London, United Kingdom; [3]School of Biological and Chemical Sciences, Queen Mary University of London, London, United Kingdom; [4]Creative Research Institution, Hokkaido University, Sapporo, Japan; [5]PRESTO, Japan Science and Technology Agency, Saitama, Japan; [6]Department of Mathematical Sciences, University of Cincinnati, Cincinnati, United States; [7]Department of Ophthalmology, College of Medicine, University of Cincinnati, Cincinnati, United States; [8]Division of Developmental Biology, Cincinnati Children's Hospital Medical Center, Cincinnati, United States

**\*For correspondence:** fumika.
hamada@cchmc.org

**Present address:** †Institute for Neuro and Behavioral Biology, Westfälische Wilhelms University, Münster, Germany

**Competing interests:** The authors declare that no competing interests exist.

**Abstract** Animals have sophisticated homeostatic controls. While mammalian body temperature fluctuates throughout the day, small ectotherms, such as *Drosophila* achieve a body temperature rhythm (BTR) through their preference of environmental temperature. Here, we demonstrate that pigment dispersing factor (PDF) neurons play an important role in setting preferred temperature before dawn. We show that small lateral ventral neurons (sLNvs), a subset of PDF neurons, activate the dorsal neurons 2 (DN2s), the main circadian clock cells that regulate temperature preference rhythm (TPR). The number of temporal contacts between sLNvs and DN2s peak before dawn. Our data suggest that the thermosensory anterior cells (ACs) likely contact sLNvs via serotonin signaling. Together, the ACs-sLNs-DN2s neural circuit regulates the proper setting of temperature preference before dawn. Given that sLNvs are important for sleep and that BTR and sleep have a close temporal relationship, our data highlight a possible neuronal interaction between body temperature and sleep regulation.

## Introduction

In humans, the phenomenon by which body temperature fluctuates by about one degree over a span of 24 hr is called body temperature rhythm (BTR). From a physiological perspective, this body temperature rhythm is related to sleep. For example, when body temperature decreases at the transition from day to night, sleep latency increases (*Refinetti and Menaker, 1992*; *Aschoff, 1983*; *Kräuchi, 2002*; *Weinert, 2010*; *Gerhart-Hines et al., 2013*; *Kräuchi, 2007a, 2007b*; *Gilbert et al., 2004*). Little is known, however, about the underlying relationships between sleep and body temperature in terms of neural regulation.

We have previously shown that *Drosophila* exhibit a daily temperature preference rhythm (TPR), in which their preferred temperatures increase during the daytime and then decrease at the

transition from day to night (night-onset) (*Kaneko et al., 2012*). Because *Drosophila* are small ecto-therms, they achieve a BTR through their choice of environmental temperature (*Stevenson, 1985*). Therefore, the concept of BTR is not restricted to mammals, and it is likely that BTR is evolutionarily conserved in mammals and *Drosophila*. Interestingly, our data suggest that TPR is regulated independently from locomotor activity rhythms because, while the dorsal neurons 2 (DN2s) function as a main clock to set the rhythmicity of TPR, PDF expressing lateral ventral neurons (LNvs) are not required for TPR during the daytime (*Kaneko et al., 2012*). On the other hand, LNvs are important for locomotor activity, but DN2s are not. Similarly, in mammals, BTR and locomotor activity rhythms are regulated by different subsets of subparaventricular zone (SPZ) neurons, suggesting that these rhythms are controlled independently as well (*Saper et al., 2005a*, *2005b*). Therefore, the mechanisms of TPR appear to be distinct from locomotor activity mechanisms and resemble mammalian BTR.

Sleep and BTR have a close temporal relationship in mammals and the mechanisms controlling sleep in flies are analogous to those in mammals (*Crocker and Sehgal, 2010*; *Hendricks et al., 2000*; *Shaw et al., 2000*). In *Drosophila*, pigment dispersing factor (PDF)-expressing small and large lateral ventral neurons (LNvs) play an important role in modulating sleep and arousal (*Sheeba et al., 2008*; *Parisky et al., 2008*) and morning anticipation in circadian locomotor activity rhythms (*Allada and Chung, 2010*; *Grima et al., 2004*; *Stoleru et al., 2004*; *Yao and Shafer, 2014*; *Stoleru et al., 2005*). Therefore, we first asked whether PDF neurons are involved in regulating TPR. We previously showed that *period* mutants ($per^{01}$) exhibit abnormal TPR during the daytime and that PERIOD expression in the PDF neurons did not restore normal TPR (*Kaneko et al., 2012*). Also, we recently showed that $Pdf^{01}$ mutants exhibit normal preferred temperature during the daytime (*Goda et al., 2016*), suggesting that neither PDF nor PDF neurons are required for regulating the rhythmicity of TPR. However, unexpectedly, we observed that $Pdf^{01}$ mutants prefer a much lower temperature late at night before dawn (ZT16-24) (*Goda et al., 2016*). Therefore, we hypothesized that PDF neurons may be involved in regulating preferred temperature before dawn.

We reveal here that sLNvs, a subset of PDF neurons, are important for pre-dawn temperature preference at ZT22-24. It has been shown that the morphology of sLNvs' projections varies across the day (*Fernández et al., 2008*; *Sivachenko et al., 2013*; *Petsakou et al., 2015*), leading to a temporal change in synaptic contacts with other neurons, which is controlled by the circadian clock (*Gorostiza et al., 2014*). Our data suggest that sLNvs contact and activate DN2s, the main clock cells for TPR (*Kaneko et al., 2012*), and that these sLNvs-DN2s contacts dramatically fluctuate during the day and peak before dawn (ZT22-24). Interestingly, our data suggest that sLNvs contact thermal sensors, anterior cells (ACs), which express Transient receptor potential (Trp) A1 and regulate temperature preference behavior (*Hamada et al., 2008*; *Tang et al., 2013*; *Galili et al., 2014*). Therefore, ACs-sLNvs-DN2s neural circuits play important roles in the regulation of pre-dawn temperature preference. Our data show a new role of sLNvs in setting the preferred temperature before dawn, suggesting a possible interaction between sleep-wake cycles and TPR at the neural level.

## Results

### PDF neurons are important for temperature preference before dawn

PDF expressing lateral ventral neurons (LNvs) play an important role in modulating sleep and arousal (*Sheeba et al., 2008*; *Parisky et al., 2008*) and morning anticipation in circadian locomotor activity rhythms (*Allada and Chung, 2010*). Based on this knowledge and our recent finding regarding the much lower temperature preference of $Pdf^{01}$ mutants before dawn (*Goda et al., 2016*), we hypothesized that PDF neurons might be involved in regulating temperature preference before dawn.

To examine the role of PDF neurons in pre-dawn temperature preference, we used flies in which PDF neurons were constitutively inhibited by overexpression of the mammalian inward rectifier $K^+$ channel *Kir2.1* (*Baines et al., 2001*). In these cells, we found that inactivated PDF neurons caused a lower temperature preference than both *Gal4/+* and *UAS/+* controls at ZT19-21 and ZT22-24 (*Figure 1A* and *Supplementary file 1*). Although *UAS-Kir2.1/+* control flies preferred a lower temperature compared to *Pdf-Gal4/+* controls (*Supplementary files 1* and *2*), the inactivation of PDF neurons caused a further significant decrease of the preferred temperature at ZT19-24 (*Figure 1A* green and blue stars, *Supplementary file 1*).

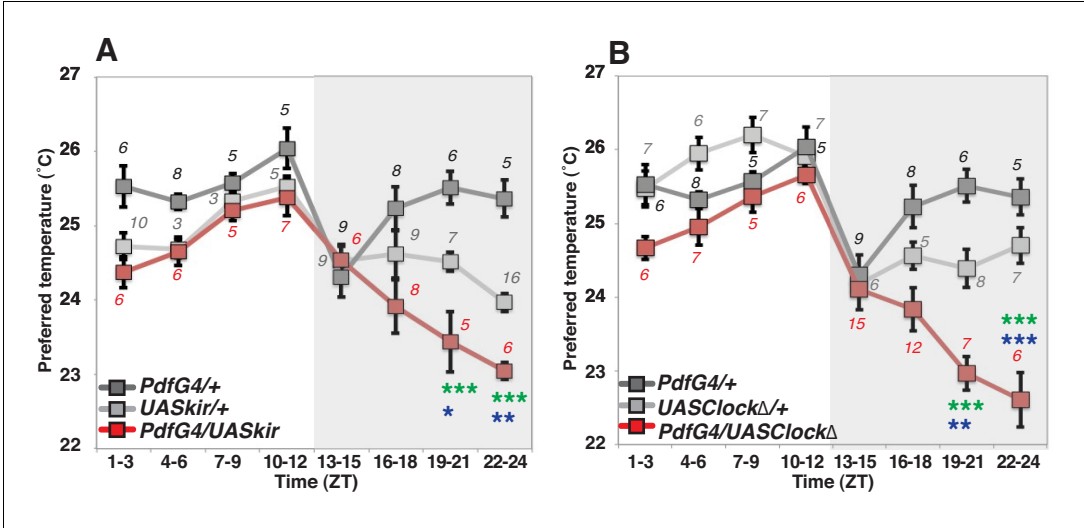

**Figure 1.** PDF neurons are required for preferred temperature before dawn. (**A**) TPR of *Pdf-Gal4/UAS-Kir2.1* (red), *Pdf-Gal4/+* (dark gray) and *UAS-Kir2.1/+* (light gray) flies over 24 hr. Numbers represent the number of assays. *Pdf-Gal4* drives sLNvs and lLNvs. (**B**) TPR of *Pdf-Gal4/UAS-ClockΔ* (red), *Pdf-Gal4/+* (dark gray) and *UAS-ClockΔ/+* (light gray) flies over 24 hr. Preferred temperatures were calculated using the distribution of flies by temperature preference behavior. Data are shown as the mean preferred temperature in each time zone (ZT1-3, 4–6, 7–9, 10–12, 13–15, 16–18, 19–21 and 22–24.) The pre-dawn time is ZT19-21 and ZT22-24. Zeitgeber Time (ZT; 12 hr light/dark cycle; ZT0 is lights-ON, ZT12 is lights-OFF). Numbers represent the number of assays. The preferred temperatures among *Gal4/UAS*, *Gal4/+* and *UAS/+* flies in each time zone were analyzed using One-way ANOVA and Tukey-Kramer tests (***Supplementary file 1***). Stars indicate p values of Tukey-Kramer tests when *Gal4/UAS* are statistically different from both *Gal4/+* (stars in green) and *UAS/+* (stars in blue). ****p<0.0001, **p<0.01 or *p<0.05.

Because PDF neurons are clock neurons, we also asked whether the clock in PDF neurons is responsible for this pre-dawn phenotype. We used flies with impaired clock function by expressing a dominant negative form of *Clock* in the PDF neurons which were previously shown to produce arrhythmic locomotor behavior (***Tanoue et al., 2004***). An impaired endogenous clock in PDF neurons caused lower temperature preferences at ZT19-24 but similar temperature preferences to controls most other times of the day (***Figure 1B***). Since these pre-dawn data are consistent with the phenotype of inactivation of PDF neurons (***Figure 1B***), we conclude that the circadian clock in PDF neurons is involved in regulating temperature preference before dawn. Importantly, neither inhibition of PDF neurons, nor impairment of clock function in these cells disrupts temperature preference during the daytime (***Supplementary file 3***), which is consistent with previous findings that PDF neurons (LNvs) are not required for TPR during the daytime (***Kaneko et al., 2012***; ***Goda et al., 2016***). Thus, we focused on the role of PDF neurons for temperature preference only before dawn but not during the daytime.

## sLNvs physically contact DN2s

It has been shown that sLNvs and DN2s contact each other in the larval brain (***Picot et al., 2009***) and that sLNvs project near DN2s in the adult brain (***Helfrich-Förster, 2003***). This raises the possibility that sLNvs and DN2s are directly connected in the adult as well. To test this, we performed a GFP Reconstitution Across Synaptic Partners (GRASP) experiment, which has been used to study synaptic connections in several animal models including *Drosophila* (***Feinberg et al., 2008***; ***Gordon and Scott, 2009***; ***Kim et al., 2011***). While neither split-GFP fragment fluoresces individually, split-GFP fragments can reconstitute fluorescence upon cell-cell contact (***Gordon and Scott, 2009***). We used *Clk9M-Gal4; Pdf-Gal80* (DN2 driver) and *Pdf-LexA* (LNvs driver) to express the split-GFP fragments, with *UAS-CD4:spGFP1-10* in DN2s and *LexAop-CD4:spGFP11* in LNvs, respectively. When flies had both split-GFP fragments (*Clk9M-Gal4::UAS-CD4:spGFP1-10; Pdf-Gal80 / Pdf-LexA::*

*LexAop-CD4:spGFP11)*, we observed scattered green fluorescence signals only in the dorsal lateral area of the brain (*Figure 2A*, *Figure 2—figure supplement 1*), but not in the sLNvs somas nor the other regions of their projections. On the other hand, there were no reconstituted GFP fluorescence signals in the control lines (no green signal in *Figure 2B and C*). These data suggest that sLNv and DN2 projections physically contact.

Accumulating evidence supports the idea that the morphology of sLNvs projections change and that sLNvs contact different neurons in a circadian fashion (*Fernández et al., 2008*; *Sivachenko et al., 2013*; *Petsakou et al., 2015*; *Gorostiza et al., 2014*), suggesting that sLNv targets and contacts vary over the course of 24 hr. Since PDF neurons are important for pre-dawn temperature preference (*Figure 1A and B*), it is possible that morphological changes in sLNv projections may influence pre-dawn temperature preference. In fact, we observed that clock disruption in PDF neurons caused abnormal preferred temperature before dawn (*Figure 1B*), which also supported the possibility that the clock in PDF neurons contributes to this time-dependent plasticity. Therefore, we examined the extent of contacts between sLNvs and DN2s (sLNvs-DN2s) through GRASP analysis over a 24 hr period in an 12 hr: 12 hr light:dark cycle. Strikingly, while the number of sLNvs-DN2s contacts decreased during the daytime (ZT1-12), they dramatically increased during the night (ZT13-24) and peaked at ZT22-24 (*Figure 2D, E and F*). These data are consistent with the recent finding that maximal axonal volume and spread of sLNvs projection are observed at ZT24 (*Petsakou et al., 2015*). In summary, our data show that the sLNvs contact the DN2s and that the number of contacts specifically peaks before dawn.

## A functional connection between sLNvs and DN2s

To examine the functional relationship of the temporally dynamic neuronal connectivity between sLNvs and DN2s, we used the mammalian ATP-gated ionotropic P2X2 receptor (*Yao et al., 2012*; *Barber et al., 2016*). We examined whether P2X2 expressing sLNvs were excited by bath-applied ATP (*Yao et al., 2012*; *Tian et al., 2009*) and whether the exogenous depolarization of sLNvs led to a cytoplasmic calcium change in DN2s. While P2X2 was expressed in sLNvs using *Pdf-LexA*, GCaMP3.0 was expressed in both sLNvs and DN2s by using *Clk9M-Gal4*.

GCaMP fluorescence in both sLNvs and DN2s increased upon the bath application of 3 mM ATP (upper line, *Figure 3A1 and B1*). In the negative controls, the fluorescence in sLNvs and DN2s did not change in response to vehicle (bottom line, *Figure 3A1 and B1*). The average increase over the baseline (ΔF/F) in both sLNvs and DN2s in response to ATP was significant compared to the vehicle controls (*Figure 3A2 and B2*). To confirm that the increases of the GCaMP fluorescence were specific to the P2X2 expression in sLNvs, we used flies without P2X2 expression in sLNvs and found that the fluorescence in sLNvs and DN2s did not change in response to ATP (*Figure 3—figure supplement 1*). The data suggest that the excitation of sLNvs via P2X2 and the subsequent depolarization of sLNvs leads to a cytoplasmic calcium increase in DN2s. Given that the GRASP experiments showed dynamic contacts between sLNvs and DN2s (*Figure 2A*), the data suggest that sLNvs can contact and activate DN2s in a time-dependent manner.

## DN2 inhibition causes a lower temperature preference

Because the number of contacts between sLNvs and DN2s peaked before dawn (*Figure 2D*) and because sLNvs can activate DN2s (*Figure 3A and B*), we asked what would happen to temperature preference behavior if DN2s were inhibited. To examine the behavioral function of the direct neural connection between sLNvs and DN2s, we constitutively inhibited DN2s by using *Clk9M-Gal4/UAS-Kir2.1; Pdf-Gal80/+* (*Figure 3C*) as well as *Clk9M-Gal4/UAS-Kir2.1* (*Figure 3D*) flies and tested temperature preference behavior. Interestingly, we found that DN2 inhibition caused lower preferred temperatures than controls (*Clk9M-Gal4/+; Pdf-Gal80/+, Clk9M-Gal4/+* and *UAS-Kir2.1/+*) at all times of the day (*Figure 3C and D*). Because DN2 inhibition does indeed strongly cause the flies to prefer a lower temperature than controls, the data suggest that sLNvs activate DN2s to obtain a proper pre-dawn temperature preference behavior.

Although DN2s are the main clock cells for TPR (*Kaneko et al., 2012*), inhibition of DN2s only changed the set point of TPR and not the rhythmicity of TPR. To investigate this further, we looked at the expression efficiency of *Clk9M-Gal4*. We found that in more than 80% of the flies, *Clk9M-Gal4: Pdf-Gal80* (*Figure 3—figure supplement 2A and B*) and *Clk9M-Gal4* (data not shown) were

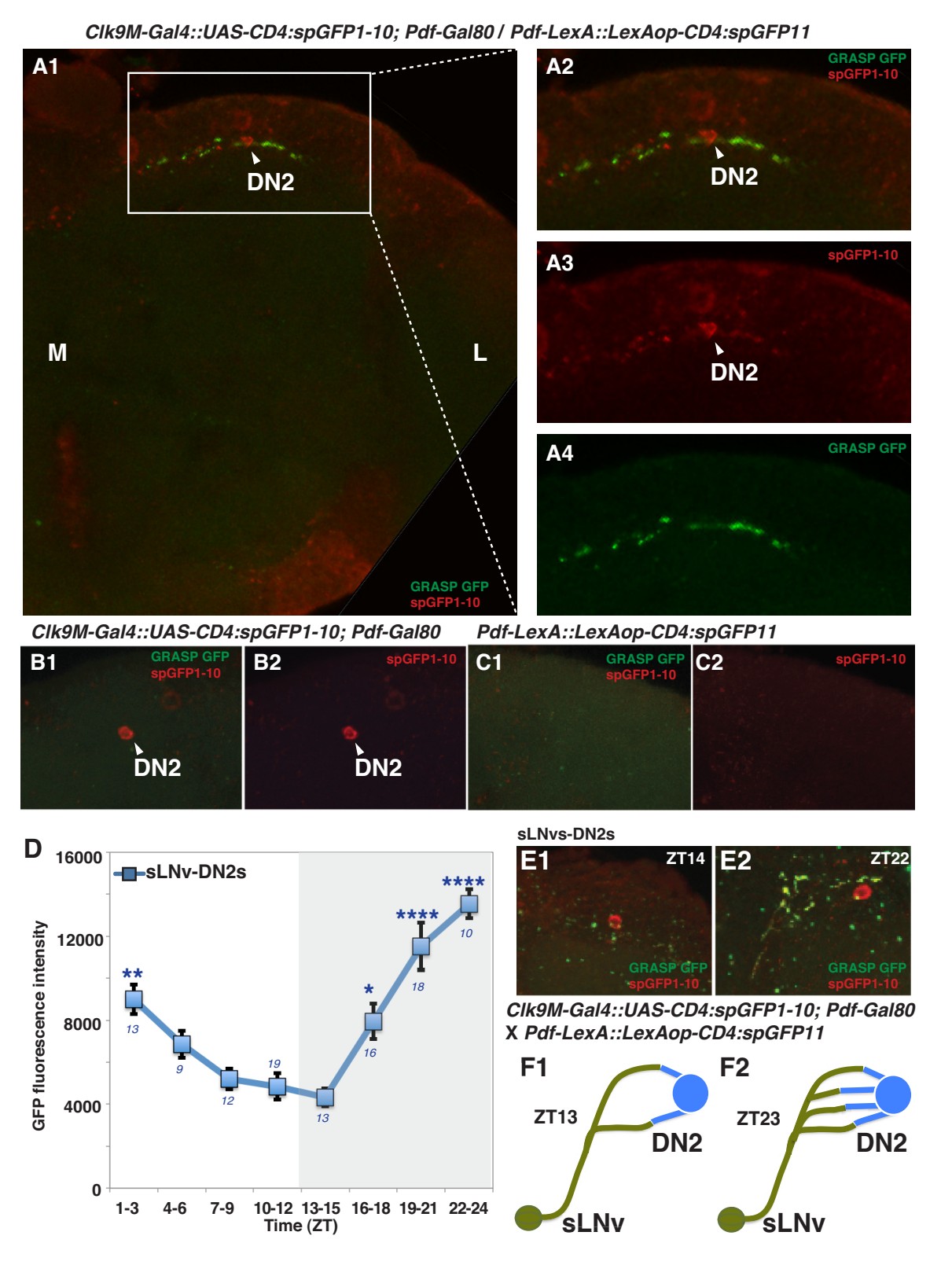

**Figure 2.** sLNvs contact DN2s, and the number of contacts peaks before dawn. (**A1**) GRASP between DN2s and sLNvs. The terminal area of sLNvs in dorsal lateral brain was magnified. The reconstituted GFP signals (green) were only detected in the dorsal protocerebrum, but not in the sLNvs soma nor the other region of their projections. *Clk9M-Gal4::UAS-CD4:spGFP1-10; Pdf-Gal80* and *Pdf-LexA::LexAop-CD4:spGFP11* flies were used to express split-GFP1-10 in DN2s, and split-GFP11 in LNvs, respectively. When the two fly lines were crossed, a reconstituted GFP signal (green) was detected

*Figure 2 continued on next page*

*Figure 2 continued*

(**A4**). The soma and projection of DN2s (red) were detected by anti-GFP (**A3**). The merged image of A3 and A4 (**A2**). M: medial, L: lateral. (**B,C**) Neither of the split GFP fragments alone in DN2s or sLNvs had reconstituted GFP fluorescence signals (no green signal in B and C). (**D**) Comparisons of GRASP signals between sLNvs and DN2s (blue) throughout the course of the day. GFP fluorescence intensity was measured and the mean values were plotted. Numbers represent the number of GRASP experiments. One-way ANOVA and Tukey-Kramer tests compared GFP fluorescence intensity of each time zone in sLNv-DN2s (blue). The intensity of all the time zones were compared with ZT 13–15 (blue). One-way ANOVA; p<0.0001, F (7, 72)=5.226. One-way ANOVA; ****p<0.0001, F (7, 102)=16.09. (**E**) A representative image of GRASP between sLNvs and DN2s at ZT14 (**E1**) and ZT22 (**E2**). *Clk9M-Gal4:: UAS-CD4:spGFP1-10; Pdf-Gal80* and *Pdf-LexA::LexAop-CD4:spGFP11* flies were used to express split-GFP1-10 in DN2s and split-GFP11 in LNvs, respectively. (**F**) A schematic of the relationship between sLNvs and DN2s at ZT13 and ZT23. The number of the sLNv-DN2 contacts dramatically fluctuates throughout the day and peak before dawn (ZT22-24) (*Figure 2D*). At ZT22-24, there are the greatest number of contacts between sLNvs and DN2s.

The following figure supplement is available for figure 2:

**Figure supplement 1.** GRASP between DN2s and sLNvs.

expressed only in one of the two DN2s in each hemisphere. Therefore, we concluded that silencing could lower only one of the two DN2 neurons' activities, which was insufficient to eliminate a rhythmic TPR since the other DN2 neuron can still drive a rhythmic TPR.

## Temperature sensing AC neurons contact sLNvs

AC neurons are warmth sensors which control temperature preference behavior (*Hamada et al., 2008*). Interestingly, AC neurons project to the dorsal protocerebrum (*Hamada et al., 2008*; *Shih and Chiang, 2011*) where nerve fibers of clock neurons are enriched (*Helfrich-Förster, 2003*). The recent paper showed that terminal projections of ACs are overlapped with dorsal projection of sLNvs (*Das et al., 2016*), suggesting that in addition to contacting the DN2, sLNv neurons may also contact AC neurons at their respective distal termini. Therefore, we sought to examine the relationship between ACs and sLNvs by GRASP.

We used *Transient receptor potential A1* (*TrpA1*)$^{SH}$-*Gal4* (ACs driver) and *Pdf-LexA* (LNv driver) to express the split-GFP fragments with *UAS-CD4:spGFP1-10* in *TrpA1*$^{SH}$-*Gal4*$^+$ neurons and *LexAop-CD4:spGFP11* in *Pdf-LexA* expressing (*Pdf-LexA*$^+$) neurons, respectively. When flies expressed both split-GFP fragments (*TrpA1*$^{SH}$-*Gal4::UAS-CD4:spGFP1-10* / *Pdf-LexA::LexAop-CD4:spGFP11*), we observed scattered, green fluorescent signals in the dorsal area of the brain (*Figure 4A*). *spGFP1-10* expression in the *TrpA1*$^{SH}$-*Gal4*$^+$ neurons was recognized with a GFP antibody (red signal in *Figure 4A1–2 and B*). However, in the control lines, when only one of the split-GFP fragments was expressed, there was no reconstituted GFP fluorescence signal (no green signal in *Figure 4B and C*). The data indicate that *TrpA1*$^{SH}$-*Gal4*$^+$ and *Pdf-LexA*$^+$ neurons project to the dorsal protocerebrum and form contacts with each other.

While *Pdf-LexA* is expressed in both sLNvs and lLNvs, only the sLNvs project to the dorsal area where AC neurons project (*Hamada et al., 2008*; *Helfrich-Förster, 2003*). This suggests that a potential synaptic connection is likely between sLNvs and ACs. It is important to note that both input and output synapses were detected in the fiber projection of sLNv neurons in the dorsal protocerebrum (*Yasuyama and Meinertzhagen, 2010*; *Collins et al., 2012*). However, dendrites of ACs are not observed in this area of the brain (*Shih and Chiang, 2011*), indicating that AC axons project to the dorsal protocerebrum.

That said, because *TrpA1*$^{SH}$-*Gal4* is not solely expressed in ACs, we could not exclude the possibility that other *TrpA1*$^{SH}$-*Gal4* expressing neurons contact the sLNvs in the GRASP experiments. Therefore, we searched for another *Gal4* line from the MZ and NP Gal4 lines (*Tanaka et al., 2012*) that is more selectively expressed in the AC neurons. AC neurons' somas are located in the base of the antennal nerve and project to the Superior Lateral Protocerebrum (SLP) region (*Figure 4—figure supplement 1A*) (*Hamada et al., 2008*; *Tang et al., 2013*; *Ito et al., 2014*). This unique projection pattern allowed us to identify the *NP0002-Gal4* line as an AC neuron driver. Although *NP0002-Gal4* expression is weaker than that of *TrpA1*$^{SH}$-*Gal4,* we found that *NP0002-Gal4* is more selectively expressed in the AC neurons (*Figure 4—figure supplement 1B*) than *TrpA1*$^{SH}$-*Gal4* and is only

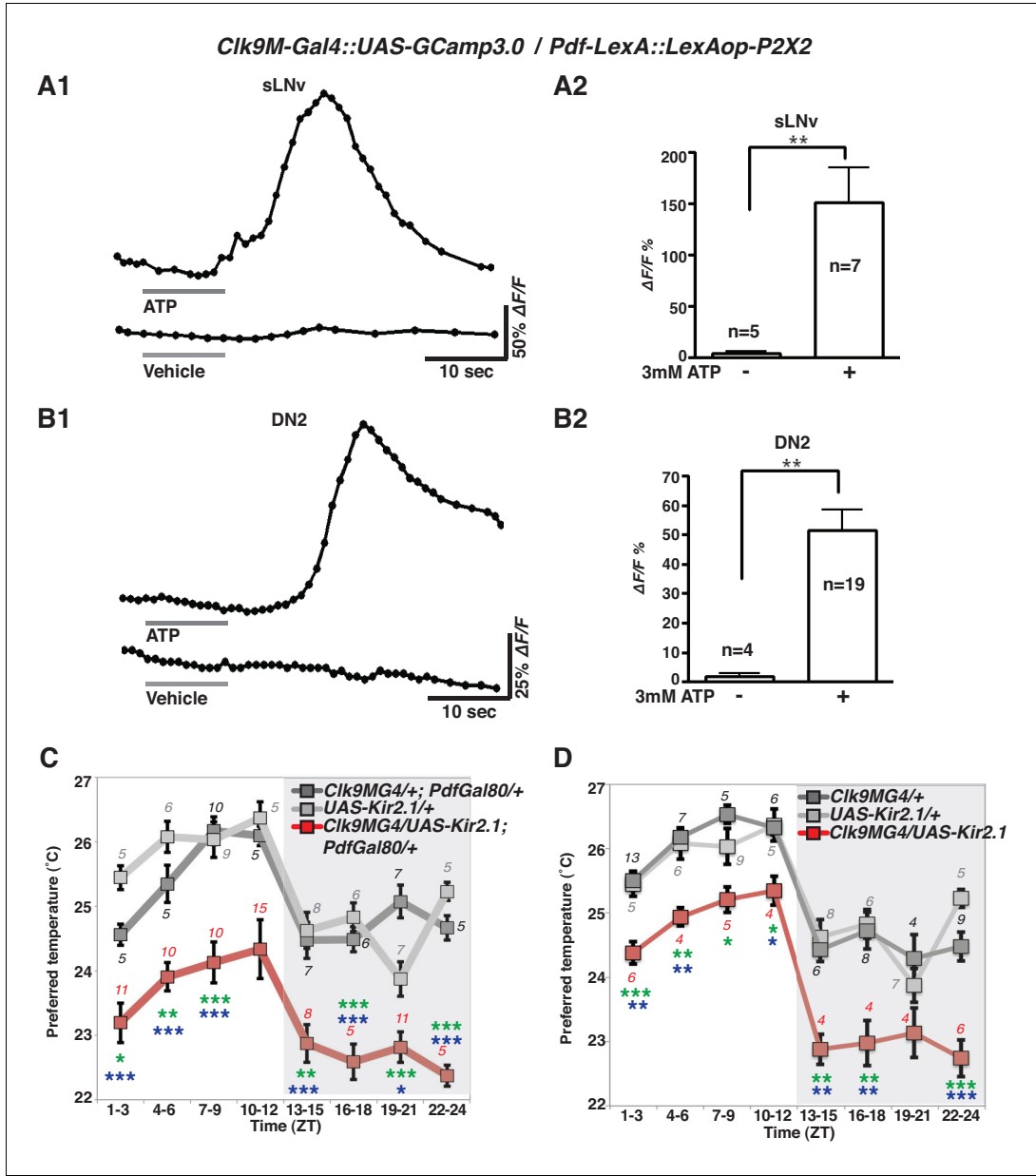

**Figure 3.** sLNvs activate DN2s, and the loss of DN2 activation results in a lower temperature preference. (**A1**) A representative graph of sLNv activation via P2X2 expression in sLNvs. Both P2X2 and GCaMP3.0 were expressed in sLNvs by using *Clk9M-Gal4::UAS-GCaMP3.0 / Pdf-LexA:: LexAop-P2X2* flies. The representative trace of GCaMP fluorescence in a sLNv neuron showed a great increase by application of 3 mM ATP into the bath (upper line), but it showed no response to the vehicle control (bottom line). ATP was present in the bath until the end of calcium imaging acquisition. (**B1**) A representative graph of DN2 activation via P2X2 expression in sLNvs. GCaMP3.0 and P2X2 were expressed in DN2s and sLNvs, respectively, by using *Clk9M-Gal4::UAS-GCaMP3.0 / Pdf-LexA:: LexAop-P2X2*. The representative trace of GCaMP fluorescence in DN2 neurons showed an excitation following the activation of P2X2 in sLNvs, which are activated by 3 mM ATP perfusion into bath (upper line), but it showed no response to the vehicle control (bottom line). (**A2, B2**) The bar graph shows mean maximum GCaMP fluorescence increase in sLNvs (**A2**) and DN2s (**B2**) to bath-applied ATP and vehicle. Unpaired t-test between ATP and vehicle. Numbers represent the number of experiments. (**C**) TPR of *Clk9M-Gal4/UAS-Kir2.1; Pdf-Gal80/+* (red), *Clk9M-Gal4/+; Pdf-Gal80/+* (dark gray) and *UAS-Kir2.1/+* (light gray) flies over 24 hr. Data are shown as the mean preferred temperature. Numbers represent the number of assays. Stars indicate p values of Tukey-Kramer tests when *Gal4/UAS* are statistically different from both *Gal4/+* (stars in green) and *UAS/+* (stars in blue). ****$p<0.0001$, **$p<0.01$ or *$p<0.05$. One-way ANOVA ZT1-3; $p=0.0002$, $F_{(2, 20)}=14.79$. ZT4-6; $p<0.0001$, $F_{(2, 20)}=21.88$. ZT7-9; $p<0.0001$, $F_{(2, 28)}=17.75$. ZT10-12; $p=0.014$, $F_{(2, 24)}=5.211$. ZT13-15; $p=0.0004$, $F_{(2, 22)}=11.99$. ZT16-18;

*Figure 3 continued on next page*

*Figure 3 continued*

p<0.0001, F (2, 16)=26.03. ZT19-21; p<0.0001, F (2, 24)=19.61. ZT22-24; p<0.0001, F (2, 14)=83.46. (D) TPR of *Clk9M-Gal4/UAS-Kir2.1* (red), *Clk9M-Gal4/+* (dark gray) and *UAS-Kir2.1/+* (light gray) flies over 24 hr. Data are shown as the mean preferred temperature. Numbers represent the number of assays. Stars indicate p values of Tukey-Kramer tests when *Gal4/UAS* are statistically different from both *Gal4/+* (stars in green) and *UAS/+* (stars in blue). ****p<0.0001, **p<0.01 or *p<0.05. One-way ANOVA ZT1-3; p=0.0005, F (2, 21)=11.36. ZT4-6; p=0.0012, F (2, 14)=11.26. ZT7-9; p=0.0163, F (2, 16)=5.387. ZT10-12; p=0.0069, F (2, 12)=7.741. ZT13-15; p=0.0012, F (2, 15) =10.91. ZT16-18; p=0.0019, F (2, 15)=9.791. ZT19-21; p=0.1172, F (2, 12)=2.577. ZT22-24; p<0.0001, F (2, 17)=24.33.

The following figure supplements are available for figure 3:

**Figure supplement 1.** Lack of P2 × 2 expression in sLNvs leads to no responses to ATP application in both sLNvs and DN2s.

**Figure supplement 2.** The DN2 driver is only expressed in one of the two sets of DN2s.

expressed in a few other cells in the Subesophageal Ganglion (SEG) (*Figure 4—figure supplement 1A2*).

In *NP0002-Gal4* (independent ACs driver) and *Pdf-LexA* flies (LNv driver), we observed a GRASP signal in the same brain region as *TrpA1^{SH}-Gal4* and *Pdf-LexA* GRASP signals, although the signals were very weak due to the lower expression of *NP0002-Gal4* (data not shown). Because *NP0002-Gal4* is expressed more specifically in AC neurons than *TrpA1^{SH}-Gal4*, these data further support that ACs and sLNvs may have synaptic connectivity.

## ACs are involved in regulating temperature preference before dawn

To examine the functional role of ACs in the regulation of temperature preference before dawn, we used the flies in which AC neurons are inhibited by Kir2.1 (*Baines et al., 2001*) using the AC neuron driver *TrpA1^{SH}-Gal4*. We observed that the inactivation of *TrpA1^{SH}-Gal4^+* neurons caused the flies to prefer significantly lower temperatures than controls at ZT19-21 and 22–24 (*Figure 4D* and *Supplementary file 1*).

Next, we used *TrpA1-RNAi* within the *TrpA1^{SH}-Gal4^+* cells. Because ACs are the only TRPA1 positive cells labeled by the TRPA1 antibody among those *TrpA1^{SH}-Gal4^+* neurons, TRPA1 in ACs was knocked down using *TrpA1^{SH}-Gal4/UAS-TrpA1-RNAi* flies (*Hamada et al., 2008*). We found that TRPA1 knockdown in ACs caused a lower temperature preference than both *Gal4* and *UAS* controls before dawn (ZT22-24) (*Figure 4—figure supplement 1C* and *Supplementary file 1*). Furthermore, TRPA1 knockdown in ACs by *NP0002-Gal4* caused a similar lower temperature preference phenotype (*Figure 4—figure supplement 1D* and *Supplementary file 1*) to that of TRPA1 knockdown in ACs by *TrpA1^{SH}-Gal4*. Therefore, these data further support the conclusion that ACs are involved in regulating temperature preference before dawn (ZT 22–24).

Additionally, we observed that the inactivation of *TrpA1^{SH}-Gal4* expressing (*TrpA1^{SH}-Gal4^+*) neurons caused a higher temperature preference than controls (*TrpA1^{SH}-Gal4/+* and *UAS-Kir2.1/+*) at ZT1-3 (*Figure 4D* and *Supplementary file 1*). However, AC neuron involvement at ZT 1–3 is still not conclusive because there was little difference of preferred temperatures between *UAS-TrpA1-RNAi/+* and *TrpA1^{SH}-Gal4/UAS-TrpA1-RNAi* flies (*Figure 4—figure supplement 1C* and *Supplementary file 3*). Particularly, *UAS-TrpA1-RNAi/+* flies exhibited an abnormal TPR phenotype with a higher preferred temperature than normal at ZT 1–3. Because there are four UAS transgenes in the *UAS-TrpA1-RNAi/+* flies, we suspect that they may have caused leaky expression of *UAS-TrpA1-RNAi*, which could have resulted in the abnormal temperature preference.

On the other hand, the inactivation of *TrpA1^{SH}-Gal4^+* neurons and the TRPA1 knockdown in ACs causes the robust and reproducible lower temperature preference phenotype mostly before dawn (ZT22-24) (*Figure 4D* and *Figure 4—figure supplement 1C–D*). Therefore, we concluded that ACs are involved in regulating temperature preference before dawn (ZT 22–24).

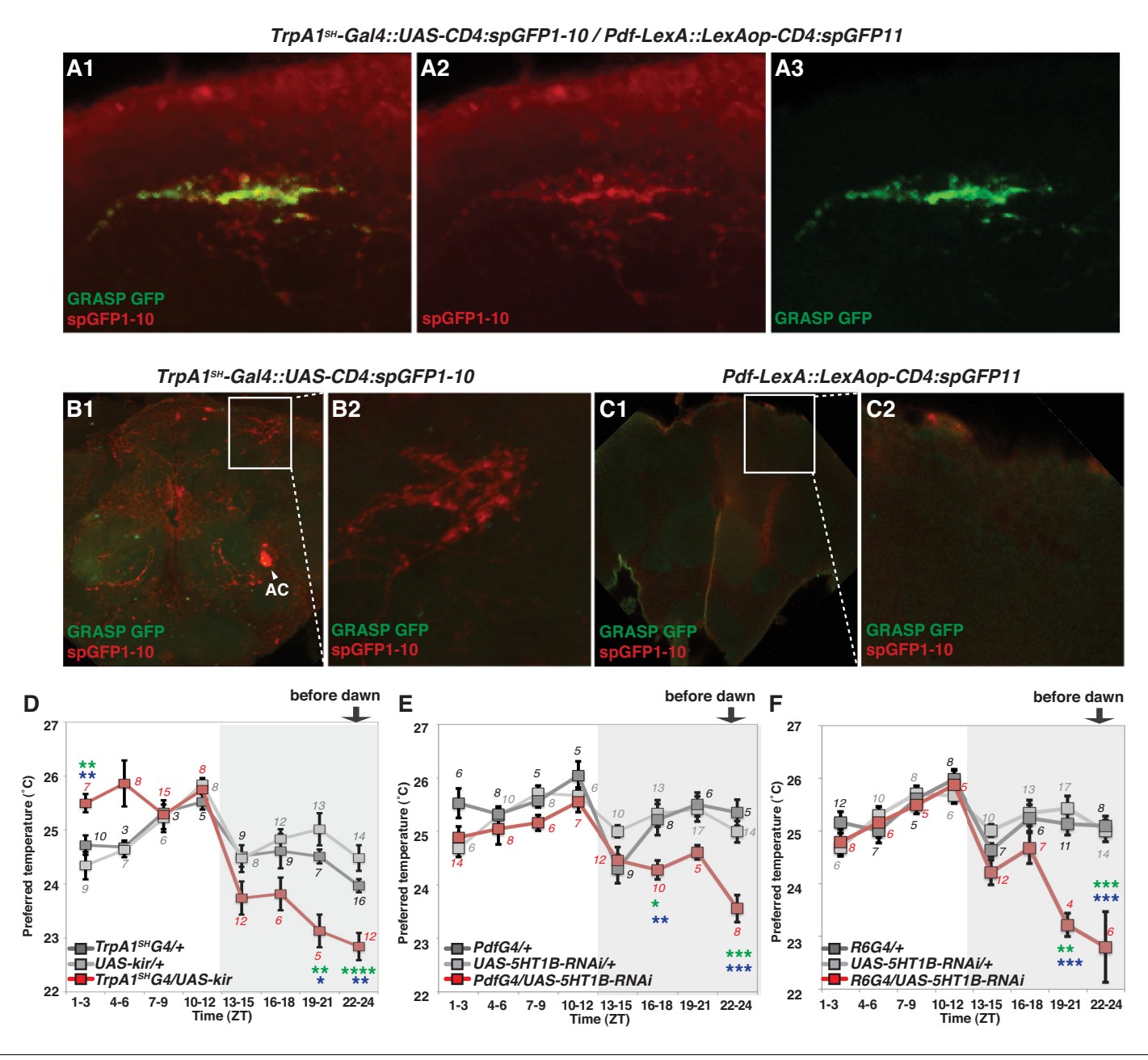

**Figure 4.** *TrpA1^{SH}-Gal4* and *Pdf-Gal4* expressing neurons contact and are involved in regulating temperature preference before dawn. (**A**) GRASP between ACs and sLNvs. *TrpA1^{SH}-Gal4::UAS-CD4:spGFP1-10* and *Pdf-LexA::LexAop-CD4:spGFP11* flies were used to express split-GFP1-10 in ACs and split-GFP11 in LNvs, respectively. When these fly lines were crossed, a reconstituted GFP signal (green) was detected only at the distal terminus (**A3**). The axons of ACs (red) were labeled by anti-GFP-Cy5 (**A2**), which can detect split-GFP1-10 expressed in AC neurons. The merged image of A2 and A3 (**A1**). (**B,C**) Neither of the split GFP fragments alone in ACs or sLNvs had a reconstituted GFP fluorescence signal (green). (**B2, C2**): magnified images of B1 and C1, respectively. (**D**) TPR of *TrpA1^{SH}-Gal4/UAS-Kir2.1* (red), *TrpA1^{SH}-Gal4/+* (dark gray) and *UAS-Kir2.1/+* (light gray) flies over 24 hr. (**E**) TPR of *Pdf-Gal4/UAS-5HT1B-RNAi* (red), *Pdf-Gal4/+* (dark gray) and *UAS-5HT1B-RNAi /+* (light gray) flies over 24 hr. (**F**) TPR of *R6-Gal4/UAS-5HT1B-RNAi* (red), *R6-Gal4/+* (dark gray) and *UAS-5HT1B-RNAi /+* (light gray) flies over 24 hr. *R6-Gal4* drives sLNvs, but not lLNvs. The preferred temperatures among *Gal4/UAS*, *Gal4/+* and *UAS/+* flies in the each time zone were analyzed using one-way ANOVA and Tukey-Kramer tests (*Supplementary file 1*). Stars indicate p values of Tukey-Kramer tests when *Gal4/UAS* are statistically different from both *Gal4/+* (stars in green) and *UAS/+* (stars in blue). ****p<0.0001, ***p<0.001, **p<0.01 or *p<0.05.

The following figure supplements are available for figure 4:

**Figure supplement 1.** TRPA1 knockdown in ACs causes lower temperature preference before dawn.

*Figure 4 continued on next page*

*Figure 4 continued*

**Figure supplement 2.** *TrpA1^SH^-Gal4* expressing cells do not overlap with clock neurons in the brain.

## 5HT1b in sLNvs is important for temperature preference before dawn

Because our data suggest that ACs-sLNvs neural circuits could contribute to temperature preference before dawn, we next considered which neurotransmitter is employed for transmission from ACs to sLNvs. ACs are serotonergic (*Shih and Chiang, 2011*), and LNvs express one of the serotonin receptors, 5HT1B, that contributes to circadian photosensitivity (*Yuan et al., 2005*). To test whether 5HT1B in sLNvs is important for temperature preference before dawn, we knocked down 5HT1B in PDF neurons and tested temperature preference behavior. 5HT1B knockdown in PDF neurons caused lower temperature preferences than the controls at ZT16-18 and ZT22-24, but these flies preferred similar temperatures as controls during the rest of the day (*Figure 4E* and *Supplementary file 1*).

Based on anatomical reasons explained earlier, sLNvs but not lLNvs should contact ACs. However, since *Pdf-Gal4* is expressed in both sLNvs and lLNvs, we verified the responsible neurons by using *R6-Gal4*, which is not expressed in lLNvs but is expressed in sLNvs and other non-clock neurons. Consistent with the phenotype of the 5HT1B knockdown using *Pdf-Gal4* (*Figure 4E*), 5HT1B knockdown using *R6-Gal4* caused lower temperature preferences before dawn at ZT19-21 and 22–24, and similar temperature preferences as controls during the rest of the day (*Figure 4F*). Therefore, our data indicate that 5HT1B expression in sLNvs is required for proper setting of temperature preference before dawn. We have shown that *Pdf-LexA^+^* and *TrpA1^SH^-Gal4^+^* neurons form contacts (*Figure 4A*) and that the inactivation of *Pdf-Gal4^+^* or *TrpA1^SH^-Gal4^+^* causes a lower temperature preference before dawn (*Figures 1A* and *4D*), all suggesting that ACs and sLNvs play an important role in temperature preference before dawn (ZT22-24). Therefore, our data suggest that the ACs-sLNvs neural circuits via serotonergic transmission likely contribute to temperature preference before dawn.

## AC neurons and 5HT1b are not required for temperature entrainment of the circadian clock

Next, we investigated if the temperature responses mediated by the AC neurons and their serotonergic communication with the sLNv clock neurons are also involved in the synchronization of the circadian clock to temperature cycles. To test this, we used flies in which the AC neurons were silenced either by expression of *Kir2.1*, or *tetanus toxin light chain* (*TNT*), or depleted of TRPA1 with *TrpA1-RNAi* using the *TrpA1^SH^-Gal4* driver. *TrpA1^SH^-Gal4 > Kir2.1*, *TrpA1^SH^-Gal4 > TNT*, *TrpA1^SH^-Gal4 >TrpA1-RNAi*, and control flies were first synchronized to 12 hr: 12 hr light dark (LD) cycles for 3 days at 20°C before being exposed to two phase-delayed 12 hr: 12 hr 20°C: 25°C temperature cycles (TC) in constant darkness (DD), each lasting for 5 days. This temperature range was chosen because the most drastic effect on TPR after AC neuron silencing was observed in the same range (*Figure 4D*). After the second TC, flies were released to constant conditions of DD and 20°C for another 5 days (*Figure 5* and *Figure 5—figure supplement 1*). During the initial LD cycle, wild type flies showed bimodal behavior, with morning and evening bouts of activity (*Wheeler et al., 1993*). In the following 2 TCs, main activity peaks occurred during the first half of the warm phase, with rapid phase adjustment after the second temperature shift. At the beginning of the final free run, the phase of the peak activity was aligned with the activity phase observed at the end of the second TC, indicating that the clock had been stably synchronized to the TC. The behavior of flies in which the AC neurons had been silenced (*TrpA1^SH^-Gal4> Kir2.1*, *TrpA1^SH^-Gal4> TNT*) or depleted of TRPA1 (*TrpA1^SH^-Gal4>TrpA1-RNAi*) seemed very similar to that of wild type and control flies, indicating that AC neurons are not required for temperature entrainment to 20°C: 25°C TC.

Because AC neuron silencing also affected TPR at higher temperatures (*Figure 4D*), we also performed the above temperature entrainment experiments in a 25°C: 29°C TC paradigm. Unfortunately, most of the flies with silenced AC neurons did not survive the experiment, presumably because of enhanced *Kir2.1* and *TNT* expression in *TrpA1^SH^-Gal4^+^* cells at the higher temperatures.

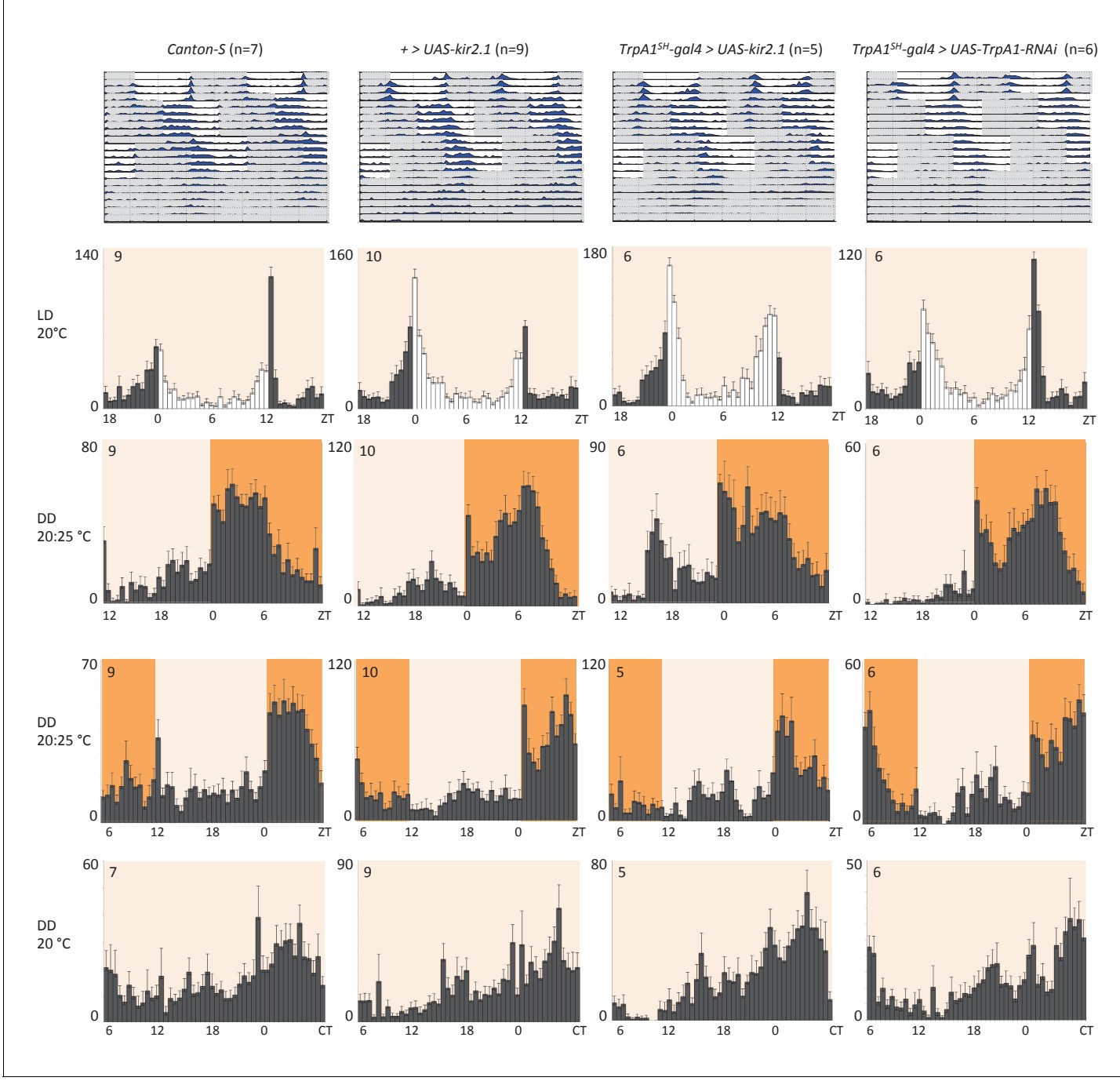

**Figure 5.** AC neuron silencing using the inwardly rectifying K[+] channel Kir2.1 and TRPA1 knock down do not interfere with temperature entrainment. Locomotor behavior of male flies of the genotypes indicated above each plot were analyzed in LD 20°C, followed by two 20°C: 25°C temperature cycles (TC) in DD, which were delayed by 6 hr compared to the previous regime. This was followed by five additional days in constant conditions (DD and 20°C). Top graphs show double-plotted average actograms, depicting behavioral activity throughout the experiment. White and grey areas depict light/warm and dark/cold periods respectively (actograms). Below, for the LD and TC parts the last 3 days and for the free-running part the first 3 days were averaged and plotted as histograms. White and grey bars indicate light and dark periods, respectively, while white background indicates 20°C periods and orange background 25°C periods (histograms). The number of animals analyzed is indicated in each histogram. The x-axis indicates time (hr) and y-axis indicates average total activity (number of beam crosses in 30 min).

The following figure supplements are available for figure 5:

*Figure 5 continued on next page*

*Figure 5 continued*

**Figure supplement 1.** Blocking synaptic transmission of AC neurons using tetanus toxin light chain (TNT) does not interfere with temperature entrainment.
**Figure supplement 2.** Silencing AC neurons does not interfere with synchronization to high temperature cycles (25°C: 29°C) (A) and down regulation of the serotonin receptor 5HT1B in *Pdf* neurons does not interfere with temperature entrainment (B).

Nevertheless, the few surviving flies showed normal entrainment to the temperature cycles (*Table 1* and *Figure 5—figure supplement 2A*), suggesting that AC neuronal activity is also not required for synchronization to 25°C: 29°C TC.

Finally, because down-regulation of the 5HT1B receptor in PDF neurons altered pre-dawn temperature preference (*Figure 4E*) and because PDF neurons are required for normal behavior during and after temperature entrainment (*Busza et al., 2007*; *Gentile et al., 2013*), we tested if *Pdf-gal4 >5HT1B-RNAi* flies showed defects in temperature entrainment. Flies were initially synchronized to 2 days of LD at 20°C before being exposed to an 8 hr advanced 20°C: 29°C TC for 5 days. After the TC, flies were released to constant conditions (DD and 20°C) for another 5–6 days (*Figure 5—figure supplement 2B*). After the temperature shift, wild type and control flies rapidly advanced their morning and evening activity peaks established during the LD entrainment. Similar to the delay experiments described above, the activity phase during the final free run was aligned with the phase established at the end of the TC, indicating that the clock had been stably entrained to the TC (*Figure 5—figure supplement 2B*). The behavior of the *Pdf-gal4 >5HT1B-RNAi* flies was indistinguishable from that of the controls. This indicates that the serotonin receptor in the PDF neurons is not required for temperature entrainment to 20°C: 29°C TCs (*Figure 5—figure supplement 2B*). In summary, ACs and the serotonin pathways of ACs-sLNvs are dispensable for temperature entrainment within the range of preferred temperatures (20°C to 29°C).

## Discussion

Our data suggest that ACs-sLNvs-DN2s neural circuits contribute to set the preferred temperature before dawn. Although sLNvs are the critical neurons for regulating the rhythmicity of the locomotor activity rhythm as well as sleep, they are not required for the rhythmicity of TPR. Instead, we found that the contribution of sLNvs in temperature preference appears to be only before dawn when animals are waking up, suggesting that sLNvs contribute to both sleep and temperature preference before dawn. Although there is a temporal relationship between BTR and sleep-wake regulation in mammals, the underlying molecular and cellular mechanisms are largely unclear. Our data raise a possible scenario in which there are some neuronal interactions between BTR and sleep-wake regulation.

**Table 1.** AC neuron silencing does not interfere with synchronization to high temperature cycles (25°C: 29°C) but reduces viability
Survival and synchronization of male flies exposed to the same temperature shift experiments as in *Figure 5*, except that the temperature cycled between 25°C and 29°C.
Number of flies that survive and entrain in 29:25 TC protocol.

| Genotype | Total *n* | Survived until end experiment *n* | Entrain of survived *n* |
|---|---|---|---|
| *Canton-S* | 10 | 6 | 6[*] |
| *TrpA1$^{SH}$-gal4 > UAS-TeTxLC V1b (inactive)* | 9 | 4 | 4 |
| *TrpA1$^{SH}$-gal4 > UAS-TeTxLC R3 (active)* | 9 | 0 | |
| *+ > UAS-kir2.1* | 10 | 8 | 8 |
| *TrpA1$^{SH}$-gal4 > UAS-kir2.1* | 10 | 1 | 1[*] |
| *TrpA1$^{SH}$-gal4 > UAS-TrpA1 RNAi* | 8 | 3 | 3[*] |

[*] actograms shown in **Figure 5—figure supplement 2A**.

### There may be a 'basal state' in temperature preference rhythm

We showed that inhibition, molecular clock disruption and 5HT1B-knockdown in PDF neurons (*Figures 1A–B* and *4E–F*), ACs' inhibition (*Figure 4D*), and DN2s' inhibition (*Figure 3C and D*) caused a lower temperature preference. However, it is unclear why these genetically manipulated flies displayed lower rather than higher temperature preference. We suspect that there may be a 'basal state' of temperature preference in flies, which could be just above that of noxious cold temperatures to avoid activation of the cold sensors. This is conceivable, given that flies shy away from noxious warm and cold temperatures to congregate at a preferred temperature. Activation of ACs-sLNvs-DN2s neural circuits before dawn may cause divergence from the basal state, while loss of ACs-sLNvs-DN2s stimulation will result in staying in the basal state with a lower temperature preference. Therefore, the clock and normal neuronal activities in sLNvs may function as a cue to prompt the flies to prefer and seek out a warmer (normal) temperature before the daytime begins.

### The neuronal plasticity of sLNvs and DN2s correlates with temperature preference behavior

We showed that the extent of sLNv-DN2 contacts fluctuate during the day (24 hr) and peaks before dawn (ZT22-24) (*Figure 2D*). Therefore, the temporal dynamics of sLNvs-DN2s connections seems to correlate with pre-dawn temperature preference behavior. Because sLNvs activate DN2s (*Figure 3A and B*) and DN2s inhibition causes a lower temperature preference (*Figure 3C and D*), a higher number of sLNv-DN2 contacts can lead to higher DN2s activation which can cause the flies to prefer a normal (warmer) temperature (*Figure 6*). On the other hand, there are fewer sLNvs-DN2s contacts during ZT4-15, which results in less DN2s modulation by sLNvs. Therefore, the temperature preference may be mainly influenced by sLNvs-DN2s before dawn. Because daily fluctuations of sLNv-DN2 contacts appear to correlate with the behavioral phenotype, sLNv-DN2 neural plasticity seems important in controlling temperature preference before dawn. Therefore, we propose a model in which sLNv-DN2 neural plasticity likely drives temperature preference before dawn.

That said, since sLNvs are important for circadian locomotor activity and are involved in arousal and clock dependent light sensitivity (*Parisky et al., 2008*; *Yuan et al., 2005*), we cannot exclude the possibility that the disrupted clock or improper activity of LNvs may change other neurons' function and thereby cause the abnormal temperature preference before dawn. Furthermore, it is also possible that sLNv-DN2 neural plasticity may just simply coincide with the pre-dawn temperature

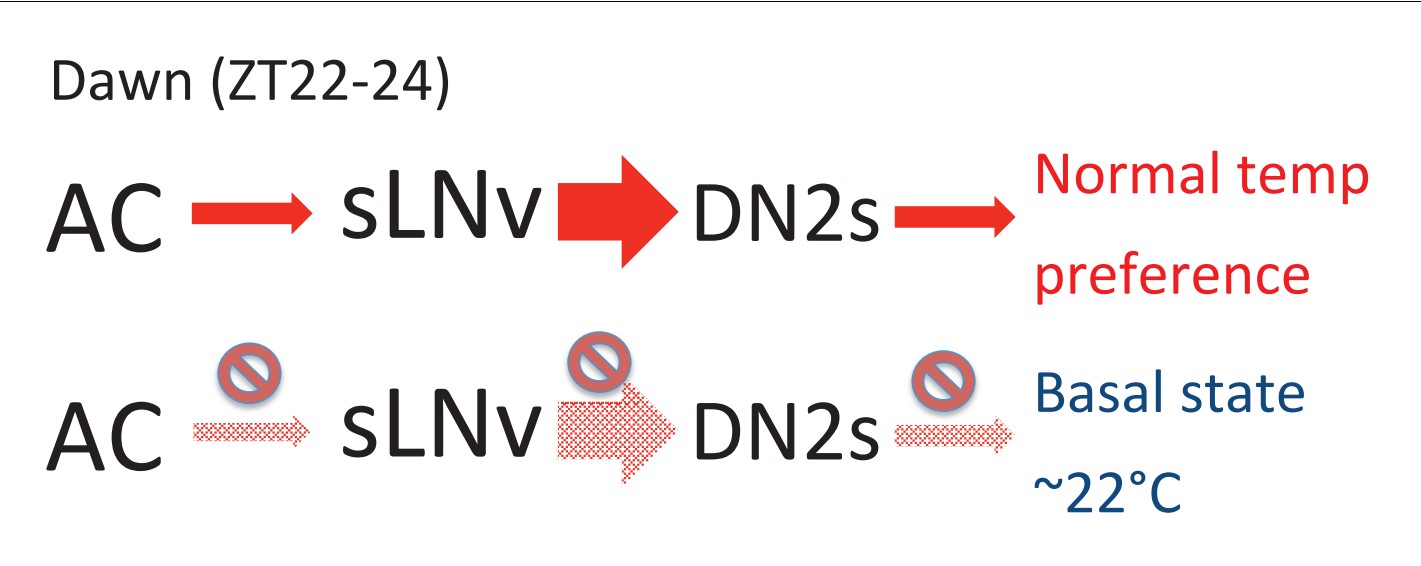

**Figure 6.** A model depicting the roles of AC, sLNv and DN2 neurons. AC neurons detect ambient temperatures, and the information could be transmitted to sLNvs. At dawn (ZT22-24) sLNvs maximally activate DN2s, and the loss of DN2 activation results in flies preferring lower temperatures (basal state). Our data suggest a likely model in which sLNvs activate DN2s before dawn to drive proper temperature preference.

preference behavior and that sLNvs and DN2s may function independently to regulate pre-dawn temperature preference.

## AC neurons are involved in regulating pre-dawn temperature preference at ZT22-24

We showed that $TrpA1^{SH}$-Gal4[+] inhibition (**Figure 4D**) and TRPA1 knockdown using $TrpA1^{SH}$-Gal4 (**Figure 4—figure supplement 1C**) or NP0002-Gal4 (**Figure 4—figure supplement 1D**) caused a lower preferred temperature at ZT22-24, suggesting that ACs are involved in regulating pre-dawn temperature preference behavior (ZT22-24). Notably, $TrpA1^{SH}$-Gal4>Kir2.1 flies (**Figure 4**) displayed a more severe phenotype than $TrpA1^{SH}$-Gal4>TrpA1-RNAi or NP0002-Gal4>TrpA1-RNAi flies (**Figure 4—figure supplement 1C–D**). Although there are many potential causes, two are particularly likely. First, it is possible that Kir2.1 has a stronger effect on AC neurons than does the TRPA1 knockdown. Second, we previously showed that AC neurons receive inputs from the TRPA channel PYREXIA (PYX) expressed in the second antennal segment (**Tang et al., 2013**). Given that ACs are activated by TRPA1 as well as by PYX (**Tang et al., 2013**), this temperature integration in ACs may contribute to the robust phenotypes of $TrpA1^{SH}$-Gal4>Kir2.1 flies and may explain the severe phenotype of neuronal silencing compared with TRPA1 knock down in the AC neurons.

Circadian clock genes are expressed throughout the body in both vertebrates and invertebrates (**Plautz et al., 1997**; **Green and Besharse, 2004**). The firing rates of sensory neurons, such as olfactory or photoreceptor neurons, are influenced by the autonomous circadian clock within those neurons (**Tanoue et al., 2004**; **Green and Besharse, 2004**; **Granados-Fuentes et al., 2006**; **Krishnan et al., 1999**). Therefore, we asked whether clock genes expressed in ACs modulate the AC neurons' activity. To examine this possibility, we performed immunostaining against the core clock protein Timeless (TIM) in AC neurons (**Figure 4—figure supplement 2**). We found that TIM was not expressed in the AC neurons, suggesting that AC neurons do not contain functional endogenous clocks. Thus, the pre-dawn temperature preference might be caused by modulation of circuits downstream of the AC neurons. If so, the ACs could transmit temperature information to the sLNvs, where the integration and modulation by the circadian clock might occur. However, we have not successfully detected any GCaMP3.0 fluorescence changes at the synaptic contacts between ACs and sLNvs or in the somas of sLNvs so far. Because ACs are serotonergic (**Shih and Chiang, 2011**) and sLNvs express the serotonin receptor 5HT1B (**Yuan et al., 2005**) (**Figure 2F and G**), driving the Epac-cAMP sensor in sLNvs may be an alternative option for detecting a functional connectivity between sLNvs and ACs. Additionally, Arclight or GCaMP6.0 might be other options. Thus, further studies are necessary for investigating the detailed properties of the functional connection between sLNvs and ACs.

Another possibility regarding ACs' involvement in the pre-dawn temperature preference is that ACs may also project to the dorsal protocerebrum where DN2s are located (**Figure 4-Figure Supplement 2C4–6**). It is also possible that ACs may directly contact DN2s and modulate the activity of DN2s. Notably, DN2s project to the Pars Intercerebralis (PI) region, which is similar to the mammalian hypothalamus (**de Velasco et al., 2007**) and is involved in metabolism, sleep, and locomotion (**Cavanaugh et al., 2014**). Therefore, ambient temperature information provided via ACs may be an important cue for a variety of clock- dependent behaviors or other physiological functions influenced by temperature. Thus, although our data suggest that the neural circuits (ACs-sLNvs-DN2s) regulate temperature preference before dawn, further studies will be needed to understand how the circuits regulate temperature preference in a time-dependent manner.

## ACs-sLNvs circuits are not required for temperature entrainment

Our data suggest that ACs-sLNvs circuits are important for pre-dawn temperature preference (**Figure 6**), raising the question of whether temperature information is being incorporated into circadian circuits. Because the circadian clock is entrained by light-dark and temperature cycles (**Busza et al., 2007**; **Glaser and Stanewsky, 2005**; **Miyasako et al., 2007**; **Yoshii et al., 2010**), it is possible that ambient temperature information via ACs is used for temperature entrainment. Although TRPA1 itself is neither required for temperature entrainment (**Das et al., 2016**; **Roessingh et al., 2015**) nor temperature compensation (**Table 2**), TRPA1 expression in $TrpA1^{SH}$-Gal4[+] neurons is required for exhibiting a 'siesta' during temperature cycles (**Das et al., 2016**; **Roessingh et al., 2015**). Because

**Table 2.** Rhythm and temperature compensation analysis of control and *TrpA1* loss-of-function mutant flies under free running conditions at different ambient temperatures.

| Genotype | Free run | n | % Rh | Period (h) ± SEM | RS ± SEM | Q10 |
|---|---|---|---|---|---|---|
| +/+ | 16°C | 9 | 64 | 24.67 ± 0.56 | 2.6 ± 0.20 | 0.97 |
| trpA1[1]/+ | 16°C | 8 | 57 | 24.22 ± 0.57 | 2.9 ± 0.24 | 0.99 |
| +/trpA1[w903*] | 16°C | 12 | 75 | 23.83 ± 0.25 | 2.9 ± 0.28 | 1.01 |
| +/Df(3L)ED4415 | 16°C | 15 | 94 | 23.85 ± 0.19 | 3.2 ± 0.19 | 1.00 |
| trpA1[1]/trpA1[1] | 16°C | 6 | 55 | 23.29 ± 0.38 | 2.0 ± 0.19 | 1.02 |
| trpA1[1]/trpA1[w903*] | 16°C | 13 | 81 | 24.23 ± 0.17 | 3.4 ± 0.27 | 1.01 |
| trpA1[1]/Df(3L)ED4415 | 16°C | 14 | 88 | 23.54 ± 0.25 | 2.9 ± 0.19 | 1.02 |
| +/+ | 29°C | 16 | 100 | 23.75 ± 0.06 | 4.8 ± 0.24 | |
| trpA1[1]/+ | 29°C | 13 | 93 | 23.75 ± 0.12 | 3.9 ± 0.31 | |
| +/trpA1[w903*] | 29°C | 16 | 100 | 24.16 ± 0.09 | 3.5 ± 0.25 | |
| +/Df(3L)ED4415 | 29°C | 11 | 79 | 23.98 ± 0.07 | 4.1 ± 0.36 | |
| trpA1[1]/trpA1[1] | 29°C | 7 | 78 | 23.57 ± 0.46 | 2.6 ± 0.26 | |
| trpA1[1]/trpA1[w903*] | 29°C | 13 | 93 | 24.56 ± 0.11 | 3.3 ± 0.26 | |
| trpA1[1]/Df(3L)ED4415 | 29°C | 14 | 88 | 24.25 ± 0.08 | 3.9 ± 0.31 | |

AC neurons also integrate the temperature information from *Pyx-Gal4* expressing neurons and PYX has a function in synchronizing the circadian clock to temperature cycles (*Tang et al., 2013*; *Wolfgang et al., 2013*), it is possible that AC neurons influence temperature entrainment. Therefore, we sought to determine whether ACs, as well as the serotonin pathways of ACs-sLNvs circuits, are involved in temperature entrainment.

We tested *TrpA1[SH]-Gal4>kir2.1* (and *TrpA1[SH]-Gal4>TNT and TrpA1[SH]-Gal4>TrpA1-RNAi*) and *Pdf-Gal4 >5HT1B-RNAi* flies in several temperature cycles. However, these flies exhibited normal temperature entrainment, indicating that the ACs and the serotonin pathways of ACs-sLNvs are not required for temperature entrainment within the range of preferred temperatures (20°C to 29°C) (*Figure 5* and *Figure 5—figure supplements 1–2*, *Table 1*). Therefore, temperature entrainment and TPR are controlled by different neural circuits.

## Is ambient temperature important for pre-dawn temperature preference associated with arousal?

We previously demonstrated that the *Drosophila* TPR resembles the mammalian body temperature rhythms (BTR) and proposed that the mechanism controlling TPR and BTR might be similar (*Kaneko et al., 2012*). Additionally, we showed that the serotonin pathway via 5HT1B is critical for temperature preference before dawn. Importantly, since serotonin also modulates body temperature in annelids and mice (*Inoue et al., 2014*; *Ray et al., 2011*), the function of serotonin in temperature regulation may also be an additional piece of evidence of the evolutional conservation from invertebrates to mammals.

In *Drosophila*, it has been shown that ambient temperature affects sleep length (*Ishimoto et al., 2012*; *Parisky et al., 2016*). In humans, the ambient temperature strongly affects the quality of sleep (i.e., too cold or too warm of an environment affects people's sleep) (*Haskell et al., 1981*; *Mallick and Kumar, 2012*). Given that ambient temperature during the night has a great impact on sleep in flies and humans, ambient temperature information before dawn may be the universal cue as an arousing signal to modulate body temperature.

## Materials and methods

### Fly lines

All flies were raised in 12 hr light/dark cycles at 25°C; Zeitgeber Time (ZT) 0 refers to lights-on, and ZT12 refers to lights-off. All fly lines used in this study were received from Bloomington Drosophila Stock Center except for the following lines: *UAS-CD4:spGFP1-10* and *LexAop-CD4:spGFP* (from Dr. Kristin Scott), *Pdf-LexA* and *LexAop-P2X2* (from Dr. Orie Shafer), *UAS-5HT1B-RNAi* (from Dr. Amita Sehgal), and *UAS-TeTxLc V1b* (inactive tetanus toxin: UAS-IMP-TNT) and *UAS-TeTxLc G* and *R3* (Active tetanus toxin: UAS-TNT (from Dr. Sean Sweeney), and *trpA1^{W903*}* (from Dr. Daniel Tracey). While *Clk9M-Gal4* is expressed in both sLNvs and DN2s, *Clk9M-Gal4; Pdf-Gal80* is expressed in DN2s (*Kaneko et al., 2012*; *Goda et al., 2016*).

It is important to note that we used *TrpA1^{SH}-Gal4* flies (*Hamada et al., 2008*). There are three sets of TRPA1 expressing neurons (Anterior cells (ACs), Lateral cells (LCs) and Ventral cells (VCs)) in the brain. *TrpA1^{SH}-Gal4* is only expressed in the AC neurons and not in LCs or VCs (*Hamada et al., 2008*). *TrpA1^{SH}-Gal4* flies were made differently from *trpA1^{Gal4}* flies (*Lee and Montell, 2013*). It is reported that *trpA1^{Gal4}* is expressed in several clock cells including, LPNs, LNvs, LNds, DN1s, DN2s and DN3s. Although two papers using *TrpA1^{SH}-Gal4* showed that *TrpA1^{SH}-Gal4* weakly overlapped with clock neurons of LNds, fifth-sLNvs and DN1a (*Yoshii et al., 2015*; *Das et al., 2015*), we did not observe that *TrpA1^{SH}-Gal4* was expressed in any clock cells (*Figure 4—figure supplement 2*). Therefore, *TrpA1^{SH}-Gal4* may be weakly expressed in some of the clock neurons, and it is not clear if they express TRPA1 endogenously.

The rhythmic TPR phenotypes of all fly lines that were used in this study are summarized in *Supplementary file 3*. Many flies showed daytime TPRs that are statistically significant. *UAS-TrpA1-RNAi/+* flies showed abnormal TPR.

### Immunohistochemistry

Immunostaining was performed as described previously (*Hamada et al., 2008*; *Tang et al., 2013*) except 10% fetal bovine serum in PBST (PBS plus 0.5% Triton X-100) was used for blocking and antibody incubations. Antibodies were used at the following dilutions: rat anti-dTrpA1 (1:1000; from Dr. Garrity), rat anti-Tim (1:200; from Dr. Rosbash), rabbit anti-GFP (1:200; Invitrogen, Cat# A6455; RRID:AB_221570), guinea pig anti-VRI (1:200, from Dr. Hardin), mouse anti-Pdf (1:200; Developmental Studies Hybridoma Bank; RRID:AB_760350), donkey anti-guinea pig Alexa Fluor 647 (1:200; Jackson Immuno Research, Cat# 706-605-148; RRID:AB_2340476), goat anti-rabbit FITC (1:200; Jackson Immuno Research, Cat# 111-095-144; RRID:AB_2337978), donkey anti-rabbit Cy5 (1:200; Jackson Immuno Research, Cat# 711-175-152; RRID:AB_2340607) and goat anti-rat Cy5 (1:200; Jackson Immuno Research, Cat# 112-175-167; RRID:AB_2338264). Mounted brains were scanned using a Zeiss LSM5 Pascal confocal microscope. Images are digitally projected Z-stacks. For GRASP experiments: the native fluorescence of reconstituted GFP was detected without antibody staining, and they are specified by overlapping with the anti-spGFP1-10 staining conjugated with Cy5 (red) in the target synaptic areas. For comparison of GRASP signals at different times of the day (*Figure 2D*), all the brain imaging was acquired with constant scanning settings. Imaris was used to quantify the intensity of reconstituted GFP signals. After background subtraction, a wide area in the dorsal brain including all the terminals of both pre- and post-neurons were measured. Inside those areas, the total intensity of all the reconstituted GFP signals were measured as an indicator of sLNv-DN2 contacts.

For VRI staining (*Figure 3—figure supplement 1B*): the mean fluorescent intensity in DN2s' somas was measured using ImageJ. Non-clock cell regions next to DN2s were selected for background subtraction for each brain sample. All the brain imaging of VRI staining was acquired with same scanning settings for comparison between different times of the day.

### GCaMP imaging

Calcium imaging was obtained from *Clk9M-Gal4::UAS-GCaMP3.0 / Pdf-LexA:: LexAop-P2X2* flies with simultaneous use of the Gal4 and LexA systems. *Pdf-LexA* drove the expression of the vertebrate purinergic P2X2 receptor in LNvs, which rendered those cells sensitive to ATP. *Clk9M-Gal4* drove the expression of the GCaMP calcium indicator in sLNvs and DN2s. GCaMP signals of DN2s

and sLNvs were from different brain samples. The GCaMP imaging was not performed at a specific time of the day.

Brain preparation was performed as described previously (*Hamada et al., 2008*; *Tang et al., 2013*). Fly brains were dissected in hemolymph-like saline (HL3) consisting of (in mM): 70 NaCl, 5 KCl, 1.5 CaCl2, 20 MgCl2, 10 NaHCO3, 5 trehalose, 115 sucrose, and 5 HEPES, pH 7.1. The prepared brain samples were mounted on a laminar flow perfusion chamber beneath the $\times$40 water immersion objective of a fixed-stage upright microscope (Zeiss Axio Examiner. Z1). For better accessibility, the brain was faced up for acquiring the sLNv signals, while the brain was faced down for acquiring the DN2 signals. During the experiments, bath application of ATP was used to activate P2X2 expressing cells. 3 mM ATP was perfused into the bath solution for ~10 s and remain in the bath until the end of calcium imaging acquiring. The fluorescence signal was continuously monitored for at least 1 min after ATP perfusion into the bath. The GCaMP fluorescence in either sLNv or DN2 neurons showed an increase in response to the bath application of ATP within the first 30 s, but showed no response to the vehicle control.

Optical images of the preparation were acquired using a digital CCD camera (C10600-10B-H; Hamamatsu) with 512 $\times$ 512 pixel resolution. Each image's data were digitized and analyzed using AxonVision 4.8.1 (Zeiss). For analysis, the mean fluorescent intensity of the monitored neuron was calculated for each frame. Concurrently, the background fluorescence (calculated from the average fluorescence of two randomly chosen non-GCaMP expressing areas) was subtracted from the mean fluorescent intensity of the regions of interest for each frame. Background-subtracted values were then expressed as percentage $\Delta F/F$, where $F$ is the mean fluorescence intensity before stimulation.

## Temperature preference behavioral assay and data analysis

Because temperature preference sometimes varies among different fly lines (*Head et al., 2015*), the temperature preference of *Gal4/UAS* flies is always compared with the *Gal4/+* or *UAS/+* controls. Furthermore, *TrpA1$^{SH}$-Gal4* and *Pdf-Gal4* are of *y w* background. In order to minimize genetic background issues, we used male *TrpA1$^{SH}$-Gal4* and *Pdf-Gal4* to cross with UAS flies.

The temperature preference behavioral assay (*Hamada et al., 2008*) was modified. The circadian clock and light affect temperature preference behavior (*Kaneko et al., 2012*; *Head et al., 2015*). Because light affects temperature preference (*Kaneko et al., 2012*; *Head et al., 2015*), the neural circuits of TPR are expected to be different between LD and DD. Therefore, we only focused on LD in this paper. The current behavioral apparatus and detailed conditions are described in (*Goda et al., 2014*). The flies were raised at 25°C in 12 hr light and 12 hr dark (LD) conditions. The temperature preference behavior was performed for 30 min in light during the daytime and dark during the nighttime in an environmental room maintained at 25°C/65–70%RH. The flies used for the behavioral assay were never reused.

The method used to calculate the mean preferred temperature has been described previously (*Kaneko et al., 2012*; *Goda et al., 2014*). After the 30 min behavioral assay, the number of flies that were completely on the apparatus was counted. Flies that were partially or completely on the walls of the apparatus cover were not counted or included in the data analysis. The percentage of flies within each one-degree temperature interval on the apparatus was calculated by dividing the number of flies within each one-degree interval by the total number of flies on the apparatus. The location of each one-degree interval was determined by measuring the temperature at six different points on both the top and the bottom of the apparatus. Data points were plotted as a percentage of flies within a one-degree temperature interval. The weighted mean preferred temperature was calculated by summing the products of the percentage of flies within a one-degree temperature interval and the corresponding temperature (e.g., fractional number of flies X 18.5°C + fractional number of flies X 19.5°C + ........ fractional number of flies X 32.5°C). We tested the temperature preference behavioral assay at least five times in each time zone (ZT1-3, 4–6, 7–9, 10–12, 13–15, 16–18, 19–21 and 22–24). If the SEM of averaged preferred temperatures was not <0.3 after the five trials, we performed more trials until SEM reached <0.3. In order to have a full curve of 24 hr, at least 40 experiments were necessary. In each time zone, the weighted mean preferred temperatures from each trial were averaged together, and the SEM was calculated.

## Temperature entrainment assay

The locomotor activity of 2–3 day old individual adult males was recorded by an automated infrared beam monitoring system (Trikinetics, Waltham USA) as described previously (*Sehadova et al., 2009*). Males were initially synchronized to 2–5 LD cycles at 20°C. After the last dark period of the final day in LD, lights were left off (DD) for the rest of the experiment. For the 20°C: 25°C TC (*Figure 5* and *Figure 5—figure supplement 1*), the temperature remained at 20°C (LD entrainment temperature) for 6 hr before being increased to 25°C (effectively inducing a 6 hr delay compared with the previous LD cycle). The initial TC continued for 5 days and was then shifted by delaying the temperature increase by 6 hr, and flies were tested for resynchronization to this shifted TC for another 5 days before being released to constant conditions (DD and 20°C). For the 25°C: 29°C TC (*Figure 5—figure supplement 2A* and *Table 1*) the regime was identical except all 20°C periods were 25°C and all 25°C periods were 29°C. For the 20°C: 29°C TC (*Figure 5—figure supplement 2B*), the temperature was raised from 20°C (LD entrainment) to 29°C 4 hr after the lights went off in the last LD cycle (effectively inducing an 8 hr advance compared with the previous LD cycle), and kept in 12 hr:12 hr 20°C: 29°C TC for 5 days before being released to constant conditions (DD and 20°C). Daily average histograms were generated in Excel and actograms were plotted using the fly toolbox and MATLab software (*Levine et al., 2002a*). For the daily average histograms showing behavior during TC only, the last 3 days of each TC were averaged to avoid inclusion of transient behavior.

## Temperature compensation assay

For temperature compensation experiments (*Table 2*), *y w; ls-tim* flies were used as control flies. *trpA1$^1$* carries a deletion of the sixth transmembrane domain (*Kwon et al., 2008*). *trpA1$^{W903*}$* carries a point mutation that encodes a stop codon in the fourth transmembrane domain (*Zhong et al., 2012*). Both are *trpA1* loss-of-function alleles. *Df(3L)ED4415* carries a deficiency (210 kb) that removes the complete *trpA1* locus (*Ryder et al., 2007*).

As with the temperature entrainment assay, the locomotor activity of 2–3 day old individual adult males was recorded using the Trikinetics system (Waltham USA). Males were initially synchronized to 3–5 days of LD cycles at 25°C. After the last dark period of the final day in LD, lights were left off (DD) and the temperature was changed to either constant 16°C or 29°C for another 7 days. The free-running period during these 7 days was calculated by autocorrelation, using Matlab software and fly toolbox scripts, as previously described (*Levine et al., 2002b*). Flies were counted as rhythmic if the calculated period reflected what was observed in the actogram and if the rhythmic statistics (RS) value was 1.5 or above. For Q10 calculations we used Q10 = (R2/R1)$^{(10/T2-T1)}$, with T1 = 29, T2 = 16 and R1 and R2 being the mean period length at 29°C and 16°C respectively.

## Search other Gal4 lines which express in ACs

AC neurons' projection patterns are very unique and were used to screen 3939 strains of MZ- and NP-series Gal4 enhancer-trap strains (*Tanaka et al., 2012*). *NP0002-Gal4* is expressed in a set of neurons whose cell bodies are located in the antennal nerve with the same projection patterns as AC neurons. Immunostaining with TrpA1 antibody was performed to confirm whether *NP0002-Gal4* expressing neuron is overlapped with AC neurons (*Figure 4—figure supplement 1B*).

## Acknowledgements

We are grateful to Drs. Sean Sweeney, Kristin Scott, Orie Shafer, Dan Tracey, and Amita Sehgal, the Bloomington Drosophila fly stock center, Vienna Drosophila RNAi Center and Developmental Studies Hybridoma Bank for the fly lines. Drs. Michael Rosbash, Paul Hardin and Paul Garrity for the antibodies. Drs. Joachim Urban. Gerhard Technau, and the members of the NP consortium for GAL4 enhancer-trap strains. We thank Drs. Emi Nagoshi, Christian Hong and the Hamada lab members for their comments and advice on the manuscript. The fly toolbox and MATLab software used in this study was kindly provided by Dr. Joel Levine and modified appropriately. Any questions relating to the version used in this paper should be sent to Dr. Ralf Stanewsky, and any subsequent use of the code should cite the original papers (*Levine et al., 2002a*, *2002b*) and state whether or not it has been modified. This research was supported by a Trustee Grant, RIP funding and Charlotte R Schmidlapp Woman Scholars program from Cincinnati Children's Hospital, the March of Dimes, and an

NIH R01 grant GM107582 to FNH JST (Japan Science and Technology)/Precursory Research for Embryonic Science and Technology (PRESTO) to FNH and NKT. Work in the RS lab was supported by BBSRC grant BB/H001204/1 and the EU (INsecTIME, Integrated Training Network, FP7).

## Additional information

### Funding

| Funder | Grant reference number | Author |
|---|---|---|
| Japan Science and Technology Agency | PRESTO | Nobuaki K Tanaka Fumika N Hamada |
| Biotechnology and Biological Sciences Research Council | | Ralf Stanewsky |
| Seventh Framework Programme | | Ralf Stanewsky |
| National Institutes of Health | R01 grant GM107582 | Fumika N Hamada |
| March of Dimes Foundation | Basil O'Connor Starter Scholar Research Award | Fumika N Hamada |

The funders had no role in study design, data collection and interpretation, or the decision to submit the work for publication.

### Author contributions

XT, Conceptualization, Data curation, Formal analysis, Validation, Investigation, Writing—original draft, Project administration, Writing—review and editing, designed the research, performed and analyzed immunostaining, calcium imaging, and behavioral experiments and wrote the manuscript; SR, Data curation, Formal analysis, Validation, Investigation, Writing—original draft, Writing—review and editing, performed and analyzed behavioral experiments and wrote the manuscript; SEH, WW, Data curation, Formal analysis, Validation, performed and analyzed behavioral experiments; MLC, Data curation, Formal analysis, Validation, Writing—review and editing, performed and analyzed behavioral experiments as well as reviewed and edited the original manuscript; NKT, Resources, Data curation, Validation, performed the Gal4 screening; SS, Resources, Data curation, Formal analysis, Validation, analyzed behavioral experiments and statistics; RS, Data curation, Supervision, Funding acquisition, Investigation, Writing—original draft, Writing—review and editing, designed the research, performed analysis and interpretation of data, as well as wrote and reviewed the manuscript; FNH, Conceptualization, Resources, Data curation, Supervision, Funding acquisition, Validation, Investigation, Visualization, Writing—original draft, Writing—review and editing, designed the research, performed analysis and interpretation of data, as well as wrote and reviewed the manuscript

### Author ORCIDs

Xin Tang, http://orcid.org/0000-0003-2991-6197
Sanne Roessingh, http://orcid.org/0000-0002-5318-6369
Nobuaki K Tanaka, http://orcid.org/0000-0002-3845-0619
Ralf Stanewsky, http://orcid.org/0000-0001-8238-6864
Fumika N Hamada, http://orcid.org/0000-0002-5365-0504

## Additional files

### Supplementary files

• Supplementary file 1. One-way ANOVA and Tukey-Kramer tests showing a comparison of each genotype of flies (*Gal4/UAS*, *Gal4/+* or *UAS/+*) within the same time zone. The preferred temperatures among *Gal4/UAS*, *Gal4/+* and *UAS/+* flies were analyzed using One-way ANOVA and Tukey-Kramer tests. In each time zone, F and p values and degrees of freedom are shown. The comparison between *Gal4/UAS* and *Gal4/+*, *Gal4/UAS* and *UAS/+* as well as *Gal4/+* and *UAS/+* are shown in green, blue and red, respectively (****$p<0.0001$, ***$p<0.001$, **$p<0.01$ or *$p<0.05$). Stars are shown

in *Figures 1* and *4D–F* and *Figure 4—figure supplement 1C–D* when *Gal4/UAS* are statistically different from both *Gal4/+* (stars in green) and *UAS/+* (stars in blue). These time zones are highlighted in orange.

• Supplementary file 2. One-way ANOVA and Tukey-Kramer tests showing a comparison of each control fly line (*Pdf-Gal4/+*, *UAS-Kir/+*, *UAS-Δclock/+*, *UAS-5-HT1B-RNAi/+* and *R6-Gal4* within the same time zone The preferred temperatures among *Pdf-Gal4/+*, *UAS-Kir/+*, *UAS-Δclock/+*, *UAS-5-HT1B-RNAi/+* and *R6-Gal4* flies were analyzed using one-way ANOVA and Tukey-Kramer tests. In the each time zone, F and p values and degrees of freedom are shown. ***p<0.001, **p<0.01 or *p<0.05.

• Supplementary file 3. One-way ANOVA and Tukey-Kramer tests showing TPR comparisons of each fly line The daytime TPR of each fly line was analyzed using one-way ANOVA. The preferred temperatures of ZT1-3 were further compared to that of each time zone (ZT 4–6, 7–9 or 10–12) by Tukey-Kramer tests. F and p values and degrees of freedom are shown. ***p<0.001, **p<0.01 or *p<0.05

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
