## [Decision Letter]

Thank you for submitting your article "The role of PDF neurons in setting preferred temperature before dawn in *Drosophila*" for consideration by *eLife*. Your article has been reviewed by two peer reviewers, and the evaluation has been overseen by K VijayRaghavan as the Senior Editor and Reviewing Editor. The following individual involved in review of your submission has agreed to reveal his identity: Todd Holmes (Reviewer #2).

The reviewers have discussed the reviews with one another and the Reviewing Editor has drafted this decision to help you prepare a revised submission.

Summary:

This works follows excellent work from the same lab on *Drosophila*'s temperature preference rhythm (TPR), which produces a daily rhythm in body temperature. The fact that ectotherms behaviorally produce such daily rhythms in body temperature suggest that such rhythms are a fundamental adaptation among animals to a rhythmic environment. Previous work had identified the DN2 class of clock neurons as the critical clock node for TPR and implicated DH31 modulation of these neurons through the Pigment Dispersing Factor Receptor PDFR as an important signaling mechanism for the normal switch to cooler preferred temperatures at night. Remarkably, the PDF peptide did not appear to be necessary for this switch but mutants lacking PDF preferred significantly lower temperatures than wild-type flies late at night. The manuscript under consideration investigates the network mechanisms by which PDF sets late night temperature preference. The manuscript provides evidence for a network, consisting of heat sensitive AC neurons modulating the s-LNvs, which form excitatory synapses on the DN2s, that governs late night temperature preference. Establishment of such a network would be a significant contribution to the field.

Essential revisions:

There are several technical issues that dampen our enthusiasm for this study as currently presented. Yet, addressing these concerns would significantly increase the confidence with which several central results could be interpreted and these concerns are addressable.

One major technical concern is the genetic drivers used to differentially express transgenes within the DN2s and s-LNvs for the GRASP and *Kir2.1* experiments. Ideally these experiments would employ completely independent GAL4 and LexA drivers, one expressing strongly in the DN2s and the other in the LNvs. Of course, life is never this easy. The GAL4 driver used here for the DN2s (*Clk9M-Gal4*) is also, unfortunately, expressed in the s-LNvs, thus requiring the need for *Pdf-GAL80* to turn-off GAL4/UAS-mediated expression in the s-LNvs. As designed, the experiments in the study essentially require *Pdf-GAL80* silencing of UAS expression to be perfectly complete. Unfortunately, there is no evidence that this is true. This leads to serious concerns about the GRASP signal seen this study and the effects of *Kir2.1* expression driven by *Clk9M-Gal4/Pdf-GAL80*, both of which may simply reflect persistent GAL4 expression in the s-LNvs.

DN2/s-LNv GRASP: The authors describe GRASP located in the dorsal protocerebrum that is with a synaptic connection between the s-LNvs and the DN2s. Maximal putative GRASP signal between s-LNvs and DN2s corresponded exactly to the previously established time of maximal spread of the s-LNv dorsal projection. Given the possibility of some GAL4 driven expression in the s-LNvs (i.e., incomplete silencing by *Pdf-GAL80*), we are worried that the authors are simply visualizing GFP reconstitution in the axoplasm of s-LNvs. How sure are they that *Pdf-GAL80* has completely silenced UAS driven spGFP1-10 expression in the s-LNvs? One way to reassure the reader (and the authors) that this isn't the case, is to show a complete lack of GRASP signal outside of the dorsal protocerebrum. It would be very reassuring, for example, to see no GFP fluorescence in the s-LNv cell bodies or the first 1/2 of their dorsal projections. We did not see this possibility addressed clearly in the Results section or the figures.

P2X2: Though the data presented for a physiological connection between s-LNvs and DN2s does support the presence of such a connection, critical negative controls for this experiment are not shown. Though the authors state that preparations in which P2X2 was not driven in the s-LNvs, displayed no responses to ATP, the details of this control were not described and data are not shown. It is critical to show the ATP responses of *Clk9M-Gal4*/UAS-GCaMP/LexAop-P2X2 flies, using the same LexAop-P2X2 element used in the experimental fly. Unfortunately, leaky P2X2 expression is always a possibility and must be controlled for. We think the authors should show these control experiments to fully control for this important observation.

*Kir2.1* inhibition of DN2s: Isn't it possible, given the concern about incomplete silencing of UAS expression by *Pdf-GAL80*, that this is simply phenocopying Pdf-GAL4/UAS-*Kir2.1* because there is enough *Kir2.1* expression in the s-LNvs to result in inhibition? This is especially worrisome in this case, as it appears that there are two copies of the *Clk9M-Gal4* present in these experimental flies.

AC/s-LNv connection. This would be a major finding with big implications for how temperature input makes its way into the circadian system. It is very nice to see that the GRASP signal between these two neuron types persists even when the more specific AC driver (*NP0002-Gal4*) is used. GRASP alone is not compelling evidence that the AC neurons form synapses on or modulatory connections with the s-LNvs. The two neurons may simply rest in close apposition (the dorsal protocerebrum is a busy place) or, if there is a connection, the PDF neurons could be modulating the AC neurons. Without evidence for a physiological connection between AC neurons and s-LNvs it is impossible to interpret the GRASP results. We are curious why the authors did not try to confirm the presence of a connection with P2X2, as this technique is used to make the case for the s-LNv to DN2 connection earlier in the paper. Existing sensors are quite likely to be sensitive enough to detect even an inhibitory or modulatory connection between the AC neurons and the s-LNvs.

In summary this is a promising and interesting study that would be significantly strengthened by an increase in technical rigor. Further, we recommend they add citation of a recent publication Barber et al., 2016, Genes & Dev, that supports their use of the pre-synaptic P2X2/post-synaptic GCaMP assay for functional connectivity.

[Editors' note: further revisions were requested prior to acceptance, as described below.]

Thank you for resubmitting your work entitled "The role of PDF neurons in setting preferred temperature before dawn in *Drosophila*" for further consideration at *eLife*. Your revised article has been favorably evaluated by K VijayRaghavan as the Senior Editor and Reviewing Editor, and two reviewers.

The manuscript has been improved but there are some remaining issues that need to be addressed before acceptance, as outlined below:

In particular, please see the concerns of reviewer 2, which needs to be well-addressed. Also, are the experiments that you suggest (re reviewer 1's) being done/done. It will not hurt to have them in place too.

*Reviewer #1:*

The authors have addressed my concern regarding the effectiveness of the *Pdf-GAL80* element in the GRASP experiments. The absence of fluorescence in the soma and initial axonal segments of the s-LNvs gives the reader confidence that the GRASP was not attained intracellularly.

The authors have also nicely addressed my technical concerns surrounding the P2X2 experiments uncovering excitatory connections between PDF expressing neurons and the DN2s. This is now a fully controlled experiment that meets the standards in the field.

There is still no direct evidence for a functional serotonergic connection between the AC neurons and the s-LNvs. But I concede that the circumstantial evidence for the modulatory connection is compelling. It's unfortunate that the authors did not push this further (they outline a reasonable set of experiments to do so in their response to the review) as it would have made this very nice study even stronger.

This is a solid contribution to the field that sheds light on the connections mediating temperature preference rhythms in *Drosophila*, a unique area of field in which the PI continues to make excellent progress.

*Reviewer #2:*

Overall, I am satisfied with the revised manuscript. However, the authors have not entirely addressed the concern of one of the reviewers about the physiological certainty of the functional connections between the AC neurons and the sLNv. Thus, I think it is in everyone's best interest if the authors revise the tone of their claims to reflect this significant qualification. At present, the language in the text is too assertive in my opinion.

---

## [Author Response]

*Essential revisions:*

*There are several technical issues that dampen our enthusiasm for this study as currently presented. Yet, addressing these concerns would significantly increase the confidence with which several central results could be interpreted and these concerns are addressable.*

*One major technical concern is the genetic drivers used to differentially express transgenes within the DN2s and s-LNvs for the GRASP and Kir2.1 experiments. Ideally these experiments would employ completely independent GAL4 and LexA drivers, one expressing strongly in the DN2s and the other in the LNvs. Of course, life is never this easy. The GAL4 driver used here for the DN2s (Clk9M-Gal4) is also, unfortunately, expressed in the s-LNvs, thus requiring the need for Pdf-GAL80 to turn-off GAL4/UAS-mediated expression in the s-LNvs. As designed, the experiments in the study essentially require Pdf-GAL80 silencing of UAS expression to be perfectly complete. Unfortunately, there is no evidence that this is true. This leads to serious concerns about the GRASP signal seen this study and the effects of Kir2.1 expression driven by Clk9M-Gal4/Pdf-GAL80, both of which may simply reflect persistent GAL4 expression in the s-LNvs.*

We recently published the immunostaining data of the Clk9M-Gal4, Pdf-GAL80>UAS-GFP flies and showed that there were no GFP signals in sLNvs (Figure 2 in Goda et al., The Journal of Neuroscience, 2016). We conclude that Pdf-GAL80 can silence the Clk9M-Gal4 expression in sLNvs and added this information in the Materials and methods.

*DN2/s-LNv GRASP: The authors describe GRASP located in the dorsal protocerebrum that is with a synaptic connection between the s-LNvs and the DN2s. Maximal putative GRASP signal between s-LNvs and DN2s corresponded exactly to the previously established time of maximal spread of the s-LNv dorsal projection. Given the possibility of some GAL4 driven expression in the s-LNvs (i.e., incomplete silencing by Pdf-GAL80), we are worried that the authors are simply visualizing GFP reconstitution in the axoplasm of s-LNvs. How sure are they that Pdf-GAL80 has completely silenced UAS driven spGFP1-10 expression in the s-LNvs? One way to reassure the reader (and the authors) that this isn't the case, is to show a complete lack of GRASP signal outside of the dorsal protocerebrum. It would be very reassuring, for example, to see no GFP fluorescence in the s-LNv cell bodies or the first 1/2 of their dorsal projections. We did not see this possibility addressed clearly in the Results section or the figures.*

We updated the GRASP image in Figure 2 to includes the area and depth from the sLNvs cell bodies to the dorsal projections. The GRASP signals were clearly visible in the dorsal protocerebrum, but not in the s-LNv cell bodies or the first half of their dorsal projections. We also confirm it using PDF antibody to label LNvs in Figure 2—figure supplement 1.

*P2X2: Though the data presented for a physiological connection between s-LNvs and DN2s does support the presence of such a connection, critical negative controls for this experiment are not shown. Though the authors state that preparations in which P2X2 was not driven in the s-LNvs, displayed no responses to ATP, the details of this control were not described and data are not shown. It is critical to show the ATP responses of Clk9M-Gal4/UAS-GCaMP/LexAop-P2X2 flies, using the same LexAop-P2X2 element used in the experimental fly. Unfortunately, leaky P2X2 expression is always a possibility and must be controlled for. We think the authors should show these control experiments to fully control for this important observation.*

As the reviewer suggested, we performed the additional negative control in the P2X2 experiment using *Clk9M-Gal4::UAS-GCaMP x LexAop-P2X2* flies (Figure 3—figure supplement 1). The fluorescence in sLNvs and DN2s did not increase upon the ATP application. On the other hand, we also performed the same experiment in Figure 3AB as a positive control and reached the same conclusion. Thus, the data indicates that the responses in Figure 3AB were specific due to P2X2 activation in sLNvs by ATP.

Kir2.1 inhibition of DN2s: Isn't it possible, given the concern about incomplete silencing of UAS expression by Pdf-GAL80, that this is simply phenocopying Pdf-GAL4/UAS-Kir2.1 because there is enough Kir2.1 expression in the s-LNvs to result in inhibition? This is especially worrisome in this case, as it appears that there are two copies of the Clk9M-Gal4 present in these experimental flies.

We would like to point out that only one copy of the *Clk9M-Gal4* was used in the *Kir2.1* inhibition of DN2s (Figure 3). Because some of the genotypes of fly lines were not clearly written in the previous manuscript, we have revised them.

As mentioned above, we showed that *Clk9M-Gal4; Pdf-GAL80* does not drive sLNvs and that sLNvs-DN2s GRASP signals were not visible in the sLNvs’ somas. The data suggest that *Pdf-GAL80* is sufficient to block the expression of *Clk9M-Gal4* in sLNvs. Therefore, we conclude that DN2 inhibition causes the phenotype of the lower temperature preference of *Clk9M-Gal4/UAS-Kir2.1; Pdf-GAL80/+* flies.

*AC/s-LNv connection. This would be a major finding with big implications for how temperature input makes its way into the circadian system. It is very nice to see that the GRASP signal between these two neuron types persists even when the more specific AC driver (NP0002-Gal4) is used. GRASP alone is not compelling evidence that the AC neurons form synapses on or modulatory connections with the s-LNvs. The two neurons may simply rest in close apposition (the dorsal protocerebrum is a busy place) or, if there is a connection, the PDF neurons could be modulating the AC neurons. Without evidence for a physiological connection between AC neurons and s-LNvs it is impossible to interpret the GRASP results. We are curious why the authors did not try to confirm the presence of a connection with P2X2, as this technique is used to make the case for the s-LNv to DN2 connection earlier in the paper. Existing sensors are quite likely to be sensitive enough to detect even an inhibitory or modulatory connection between the AC neurons and the s-LNvs.*

We have tried to obtain evidence for a physiological connection between ACs and sLNvs. We focused on both the synaptic locus of sLNvs-ACs and the somas of sLNvs using a normal upright microscope. While GCaMP3.0 was expressed in sLNvs, P2X2 was expressed in ACs. We activated ACs by ATP or alternatively by warm temperature. However, we have not successfully detected any GCaMP fluorescence change in sLNvs.

The most challenging aspect was to find an optimal depth or location at the synaptic locus of sLNvs-ACs because there was no marker at the sLNvs-ACs synaptic locus. The sLNVs-ACs contacts were fewer and weaker than that of sLNvs-DN2s based on experiments relying on GRASP signals. Therefore, the G- CaMP3.0 fluorescence changes in the synaptic regions of sLNvs-ACs might be difficult to capture. We are currently setting up confocal microscopy with GaAsP which should allow us to capture weaker signals located at the deeper brain regions.

Given that serotonin signaling is important for the ACs-sLNvs communication (Figure 4), driving the Epac-cAMP sensor in sLNvs might be an option for detecting a functional connectivity within the sLNvs-ACs synaptic locus.

Furthermore, to obtain a robust fluorescence change in both terminal regions and somas of sLNvs, other reporters such as Arclight or G-CaMP6.0 might be useful. Thus, further studies will be necessary for investigating the detailed properties of the functional connection between sLNvs and ACs.

Due to the lack of physiological evidence, the functional basis between ACs and sLNvs is still not clear. It is possible that information could be modified in both direction of the AC-sLNv connection and we do not exclude the possibility that sLNvs could modulate ACs' activity. That said, it is known that ACs are serotonergic and do not have dendrite projections in the dorsal protocerebrum. Given that 5HT1B-RNAI in sLNvs caused abnormal temperature preference before dawn, we prefer a model in which the information flows from ACs to sLNvs at this point.

*In summary this is a promising and interesting study that would be significantly strengthened by an increase in technical rigor. Further, we recommend they add citation of a recent publication Barber et al., 2016, Genes & Dev, that supports their use of the pre-synaptic P2X2/post-synaptic GCaMP assay for functional connectivity.*

We added the reference. Thank you very much for this suggestion.

[Editors' note: further revisions were requested prior to acceptance, as described below.]

*The manuscript has been improved but there are some remaining issues that need to be addressed before acceptance, as outlined below:*

*In particular, please see the concerns of reviewer 2, which needs to be well-addressed.*

We have revised and tried not to overstate the explanation of the physiological connection between ACs and sLNvs.

*Also, are the experiments that you suggest (re reviewer 1's) being done/done. It will not hurt to have them in place too.*

Because we have been setting up the new system using different UAS-reporters and/or different microscopes, it may take more time to successfully detect the physiological signals between ACs and sLNvs. Therefore, we would like to pursue this question in the next report.